# Precapillary sphincters maintain perfusion in the cerebral cortex

Søren Grubb [1,5]*, Changsi Cai[1,5], Bjørn O. Hald[1,5], Lila Khennouf [1,2], Reena Prity Murmu[1], Aske G.K. Jensen [1,3], Jonas Fordsmann [1], Stefan Zambach[1] & Martin Lauritzen[1,4]*

Active nerve cells release vasodilators that increase their energy supply by dilating local blood vessels, a mechanism termed neurovascular coupling and the basis of BOLD functional neuroimaging signals. Here, we reveal a mechanism for cerebral blood flow control, a precapillary sphincter at the transition between the penetrating arteriole and first order capillary, linking blood flow in capillaries to the arteriolar inflow. The sphincters are encircled by contractile mural cells, which are capable of bidirectional control of the length and width of the enclosed vessel segment. The hemodynamic consequence is that precapillary sphincters can generate the largest changes in the cerebrovascular flow resistance of all brain vessel segments, thereby controlling capillary flow while protecting the downstream capillary bed and brain tissue from adverse pressure fluctuations. Cortical spreading depolarization constricts sphincters and causes vascular trapping of blood cells. Thus, precapillary sphincters are bottlenecks for brain capillary blood flow.

[1] Department of Neuroscience, Faculty of Health Sciences, University of Copenhagen, DK-2200 Copenhagen N, Denmark. [2] Department of Neuroscience, Physiology and Pharmacology, University College London, Gower Street, London WC1E 6BT, UK. [3] Department of Neurosciences, University of California, San Diego, CA 92093, USA. [4] Department of Clinical Neurophysiology, Rigshospitalet, 2600 Glostrup, Denmark. [5]These authors contributed equally: Søren Grubb, Changsi Cai, Bjørn O. Hald. *email: sgrubb@sund.ku.dk; mlauritz@sund.ku.dk

Neurovascular coupling (NVC) is the signaling mechanism that links neuronal activity to local increases in cerebral blood flow[1–4]. Increased $Ca^{2+}$ in neurons and astrocytes triggers the release of vasoactive compounds that dilate capillaries and penetrating arterioles (PAs) and thereby increases blood flow. The activity-induced flow increase is based on coordinated changes in vessel diameters, which are regulated by $Ca^{2+}$ fluctuations within the vascular smooth muscle cells (VSMCs) that circumscribe arteries and larger arterioles and the pericytes that ensheathe capillaries close to the PA[5–8]. PAs branch into capillary networks that supply each cortical layer with oxygen and glucose[9]. It remains unclear how this topology achieves a balanced and adequate perfusion of capillary beds along the entire cortical depth while simultaneously shielding the delicate brain tissue from the mechanical impact of pressure. Here, we reveal the structure and function of brain precapillary sphincters, which may serve to protect capillaries from high blood pressure while preserving blood supply to all bifurcations along the PA. We characterized the precapillary sphincter as a mural cell encircling an indentation of the capillary where it emerges from the PA. The sphincter cells were morphologically similar to brain pericytes, contained α-smooth muscle actin (α-SMA), and were ensheathed by structural proteins. Precapillary sphincters were mostly present at proximal bifurcations of PAs, ideally positioned to balance perfusion along the PA and to protect against arterial pressure. Though precapillary sphincters have been known for decades[10], their existence, except within the mesentery[11–13], has remained controversial[14,15]. This study provides unequivocal structural and functional evidence of brain precapillary sphincters and examines their role in NVC and during cortical spreading depolarization (CSD).

## Results

### Precapillary sphincters at proximal branch points.

We identified precapillary sphincters in mice expressing dsRed under the control of the NG2 promoter as dsRed-positive cells encircling an indentation of the capillary lumen as it emerges from the PA branch points (Fig. 1a). Precapillary sphincters were often but not always followed by a distention of the lumen, which we denoted as the bulb. The dsRed signal from the precapillary sphincter was usually brighter than dsRed signals from other mural cells on the PAs and first order capillaries, indicating high-NG2 expression, whereas the dsRed signal from the bulb region was low (Fig. 1a, b, d). We also identified precapillary sphincters and bulbs in awake mice with chronic cranial windows (Fig. 1c and Supplementary Fig. 3, $n = 4$) and anaesthetized NG2-dsRed mice with thinned skull over the barrel cortex[16] (Fig. 1b, Supplementary Fig. 2 and Supplementary Movie 1, $n = 3$ mice). Ex vivo studies revealed that the NG2-positive cells encircling the precapillary sphincter were individual cells encompassing the sphincter at the branch point and not processes of mural cells extending from the PA (Fig. 1d). Close inspection revealed a continuum of mural cell cyto-architecture from VSMC encircled pial arterioles to pericyte ensheathed capillaries (Fig. 1e) as described previously[17–19]. The mural cell encircling the sphincter stained weakly (if any) for Nissl neurotrace 500/525[20] and not for CD146[21,22], but showed robust CD13 staining (no marker was specific for pericytes, see Supplementary Fig. 7) and α-SMA expression (see below).

Having established the structure of precapillary sphincters, we examined their occurrence and localization within the cortical vascular network. In keeping with the work of Duvernoy et al.[9], we identified a range of PA subtypes (Fig. 2b) that differed in size, branching pattern, and cortical penetration. The heterogeneity in PA subtypes was partially reflected in the localization and frequency of sphincter and bulb occurrence. Out of the 108 PAs with 602 branches we could resolve in 9 mice examined, we found that 72% contained at least one sphincter (and that each PA had on average 28% branches with a sphincter). Precapillary sphincters localized predominantly in the upper layers of the cortex (Fig. 2c) and were observed mainly at the proximal PA branch points (Fig. 2d) of

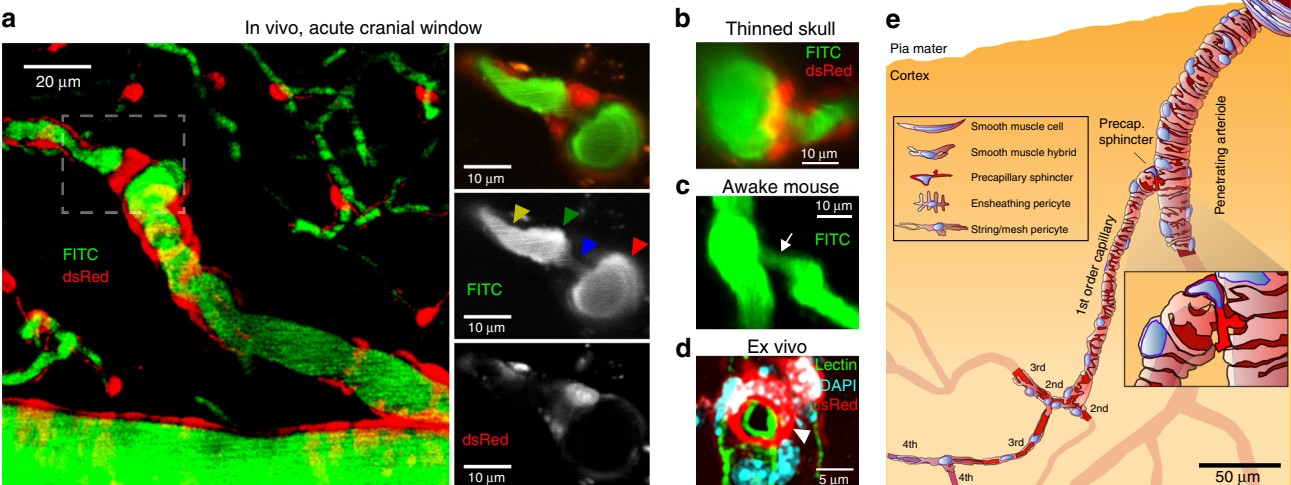

**Fig. 1 Sphincters on proximal branches of penetrating arterioles. a** Left panel: Maximal intensity projected in vivo two-photon laser scanning microscopy image of an NG2-dsRed mouse barrel cortex. An indentation of the capillary lumen is observed at the branching of the PA and is encircled by bright dsRed cell(s) (dashed insert). This structure is denoted as a precapillary sphincter. Immediately after the sphincter, a sparsely dsRed-labeled distention of the capillary lumen is observed, which we refer to as the bulb. Right panels: Single z-plane showing overlay, FITC-channel, and dsRed channel of the dashed insert. Arrows indicate the PA (red), sphincter (blue), bulb (green), and 1st order capillary (yellow). **b–d** Local TPLSM projections of precapillary sphincters in the cortex of a thinned skull mouse in vivo (**b**), an awake mouse harboring a chronic cranial window in vivo (**c**) with white arrows marking the precapillary sphincter, and an ex vivo coronal slice of a FITC-conjugated lectin (green) stained NG2-dsRed mouse (red) with DAPI-stained (blue) nuclei (**d**). The precapillary sphincter cell nucleus is arched, as it follows the cell shape, and is marked by a white arrowhead. **e** Schematic of a PA with the a a precapillary sphincter at the proximal branch point. The illustration is based on confocal imaging of coronal slices ex vivo and the exact morphology and location of NG2-dsRed positive cells and their DAPI stained nuclei are shown. For the complete figure including a venule, see Supplementary Fig. 8.

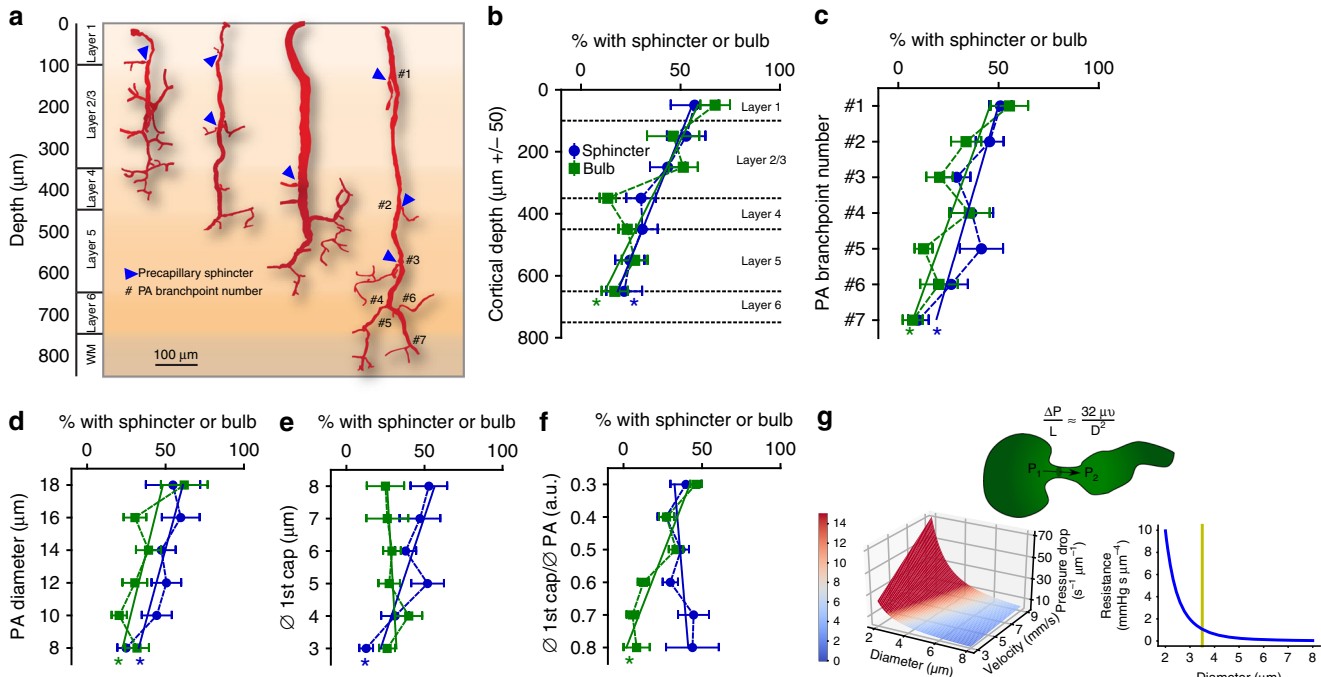

**Fig. 2 Location of sphincters help pressure equalization along PA. a** Representatives of four PA subtypes reaching different cortical layers based on ex vivo data. Precapillary sphincters are found at varying depths (marked by blue arrowheads and branchpoint numbers are indicated on the right PA). **b–f** Dependency of the presence and location of precapillary sphincters and bulbs (binned quantification) on various parameters. Criteria for the positive presence of sphincter or bulb at a branch point: sphincter <0.8 and bulb >1.25 times the diameter of a first order capillary, in total 602 branchpoints of 108 PAs in 9 mice were analyzed, ±SEM, linear regression, * = slope deviates significantly from 0. **b** Dependency on cortical depth (bin size 100 μm). **c** Dependency on PA branch number (counting from the proximal end). **d** Dependency on PA diameter (bin size 2 μm). **e** Dependency on first order capillary diameter (bin size 1 μm). **f** Dependency on first order capillary/PA diameter ratios (bin sizes as in **d** and **e**). **g** Top panel: Illustration of a pressure decrease across a precapillary sphincter and modified expression of Poiseuille's law. ΔP is the pressure difference, L unit length, μ viscosity, and υ flow velocity. Lower left: Illustration of Poiseuille's law showing how the pressure drop (defined as pressure difference per unit length times viscosity, $\frac{\Delta P}{\mu L}$, also unit of color scale), depends on the cylindrical lumen diameter and flow velocity. Note how the pressure drop increases with lumen diameters below 4 μm. Lower right: Combining flow resistance in laminar fluid flow with Poiseuille's law yields an equivalent representation of how flow resistance (defined as resistance per unit length and viscosity, $\frac{R}{\mu L}$) depends on lumen diameter. Source data are provided as a Source Data file.

relatively large PAs branching into relatively large first order capillaries (Fig. 2e, f). Thus, sphincters localize to large proximal vessels that have higher blood pressures than smaller downstream vessels. The bulb usually succeeded a sphincter but was less prevalent and did not correlate positively with the diameter of first order capillaries (Fig. 2e); bulbs were prevalent when the PA diameter was large compared to the first order capillary (Fig. 2f). For branches positive for a precapillary sphincter, the average diameter of the PA was 11.4 ± 0.6 μm, the precapillary sphincter 3.4 ± 0.2 μm, the bulb 5.8 ± 0.2 μm, and the first order capillary 5.3 ± 0.2 μm. As per Poiseuille's law (adjusted for flow velocity, Fig. 2g), a lumen diameter of 3–4 μm is at the border of high flow resistance, providing an effective means of changing the pressure drop per unit length. We conclude that precapillary sphincter complexes (sphincter and bulb) are characterized by an indentation of the lumen at the branch point encircled by a mural cell, usually followed by a distention (the bulb), and are common at proximal PA branch points of larger PAs in the mouse cortex.

**Precapillary sphincters regulate blood flow.** Having established the occurrence and morphology of precapillary sphincter complexes, we examined their role in blood flow regulation. First, we confirmed expression of α-SMA within the precapillary sphincter mural cell in coronal slices of NG2-dsRed mice (Fig. 3a, vascular lumen and cell nuclei co-stained with lectin and DAPI, respectively). Supplementary Fig. 4 and Supplementary Movie 2). Next,

we analyzed the vasomotor responses of the PA, precapillary sphincter, bulb, and first order capillary vessel segments in response to electrical whisker pad stimulation in an in vivo two-photon setup (Supplementary Fig. 1). Careful placement of linear regions of interest (ROIs) in image hyperstacks were used to avoid intersegmental interference in diameter calculations before and during whisker stimulation (Fig. 3b, c). Precapillary sphincters dilated during stimulation, followed by a poststimulus undershoot (constriction) 20–30 s after stimulation. Using four-dimensional hyperstack imaging[23], we confirmed that the undershoot was not an artifact of drift on the z-axis (Supplementary Movie 3). Relative diameter changes were significantly larger at the sphincter than the PA and the rest of the first order capillary during both dilation (33.75 ± 4.08%, Fig. 3e and Supplementary Table 1) and the undershoot (−12.40 ± 2.10%, Fig. 3f and Supplementary Table 1). To estimate the corresponding changes in flow resistance per unit length, we applied Poiseuille's law at baseline, maximal dilation and maximal undershoot (Fig. 3g–i). The flow resistance of the sphincter at rest was significantly greater than in the other segments and decreased significantly more (65.9% decrease, Fig. 3h) during dilation compared to all other segments (40.8% for the first order capillary, Fig. 3h). During the poststimulus undershoot, flow resistance increased by 80.2% at the sphincter (Fig. 3i), highlighting the sensitivity of flow resistance to sphincter constriction due to the power law relationship between diameter and flow resistance (Fig. 2g). Moreover, we observed that the length of precapillary

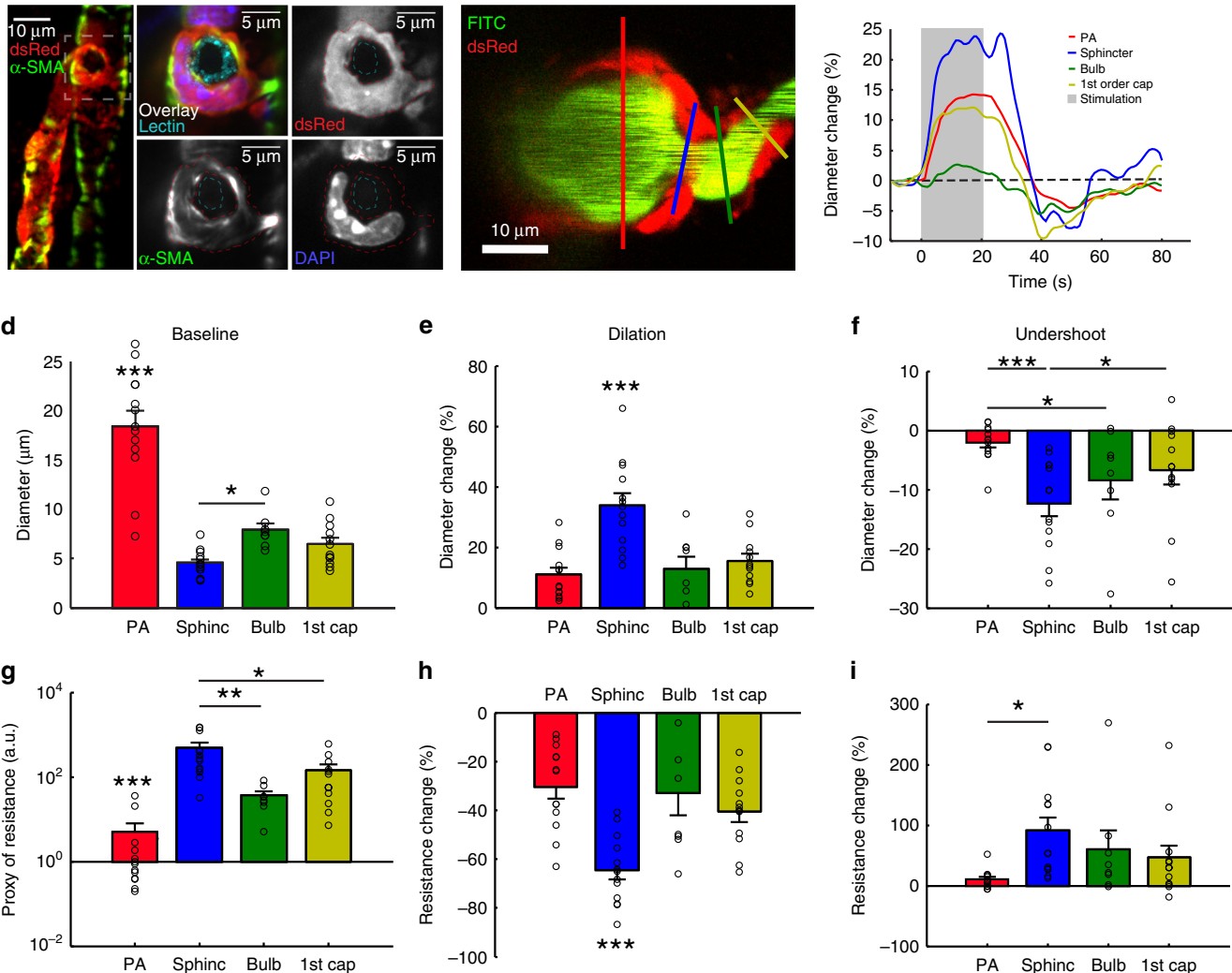

**Fig. 3 Sphincters actively regulate blood flow. a** Ex vivo coronal slices of an FITC-lectin-stained NG2-dsRed mouse immunostained for α-SMA. Left panel: maximal projection of a PA with a precapillary sphincter at the first order capillary branch point. The marked area is shown on the right. Right panels: local maximal intensity projections of the precapillary sphincter region of dsRed, α-SMA, DAPI, or all channels including FITC-lectin overlaid. The lumen (cyan) and the outlines of the dsRed signal of the precapillary sphincter cell have been marked by dashed lines in the three grayscale images. **b–i** In vivo whisker pad stimulation experiments (anaesthetized NG2-dsRed mice) using maximal intensity projected 4D data obtained by two-photon microscopy, $n = 13$ mice for PA and sphincter, 8 for bulb and 12 for first order capillary, ±SEM. **b** Maximal intensity projection of a PA branch point where the colored lines indicate the ROIs for diameter measurements of the vessel segments: PA (red), precapillary sphincter (blue), bulb (green), and first order capillary (yellow). **c** Representative time series of relative diameter dynamics in each vessel segment upon 20 s of 5 Hz whisker pad stimulation (gray bar, start at time zero). **d** Summary of baseline diameters (absolute values). **e** Summary of peak diameter change upon whisker pad stimulation. **f** Summary of the peak undershoot phase after whisker pad stimulation. **g** A proxy of flow resistance at baseline estimated using Poiseuille's law. **h** Relative change in flow resistance at peak dilation during stimulation. **i** Relative change in flow resistance during the poststimulation undershoot. The Kruskal–Wallis test was used in (**d**, **g**, and **i**) to reveal differences among vessel segments, followed by a Wilcoxon rank-sum test (with Holm's p value adjustment) for pairwise comparisons. LME models were used in (**e**, **f**, **h**, and **m**) to test for differences among segments, followed by Tukey post hoc tests for pairwise comparisons. In each figure, significance codes *$p < 0.05$, **$p << 0.01$, and ***$p < 0.001$. Source data are provided as a Source Data file.

sphincters decreased during stimulation and increased during the undershoot (Supplementary Fig. 5). Shortening of the sphincter decreases the absolute flow resistance across the precapillary sphincter and vice versa, augmenting the pressure drop reduction across the sphincter during stimulation and the pressure drop increase during the poststimulus undershoot.

Next, we examined the correlation between red blood cell (RBC) flux and diameter changes in response to whisker pad stimulation (Fig. 4a–d). RBC velocity fluctuated in synchrony with systolic and diastolic oscillations in arterial blood pressure (Fig. 4a, b). At rest, the average RBC velocity through precapillary sphincters was 8.7 ± 0.6 mm/s (Fig. 4c), significantly higher than

for the bulb (3.6 ± 0.6 mm/s) and the first order capillary (4.7 ± 0.6 mm/s), but correlated with the relative differences in the resting diameters of the vessel segments. As shown in Fig. 2g, high RBC velocity through the narrow lumen of the precapillary sphincter amplifies the reduction in pressure across the sphincter due to high shear, i.e., augments the reduction of pressure from larger proximal PAs to downstream capillaries. From the baseline measures, the pressure drop per unit length is 4-times larger in the sphincter than the first order capillary, assuming that RBC velocity and fluid velocity are equal (see Fig. 2g). During whisker stimulation (Fig. 4c), both diameter and RBC velocity increased in each segment, but significantly more at the precapillary

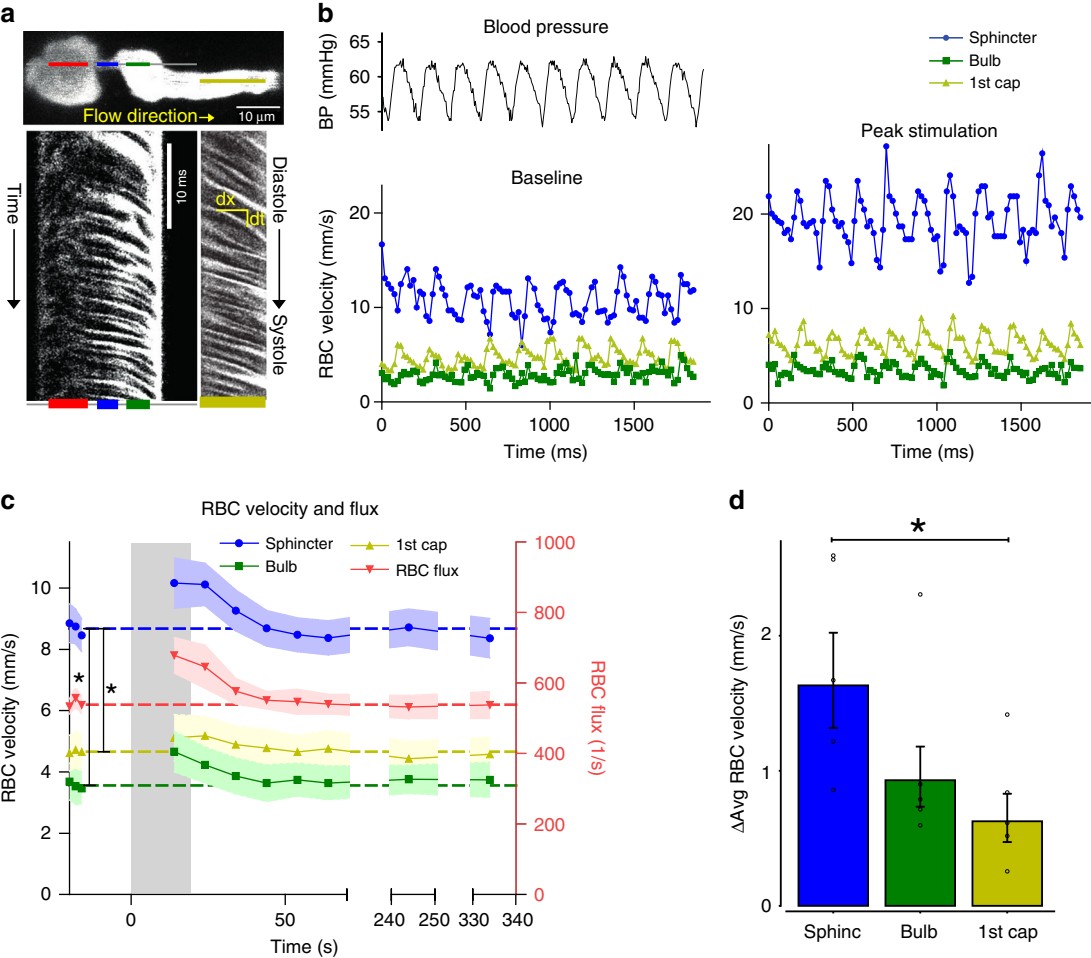

**Fig. 4 Red blood cell velocity and flux at the sphincter. a** Resonance scanning allows for rapid repetitive line-scans in a single z-plane (upper panel). In the resulting space–time maps (lower panel), individual cells appear in black with an angle proportional to the cell velocity. Red, blue, green, and yellow lines indicate the regions of the line-scans derived from the PA, sphincter, bulb, and first order capillary (first order capillaries were mostly scanned in consecutive experiments). **b** Fluctuations in femoral artery blood pressure (left upper panel) and RBC velocity (left lower panel) correlated. During whisker pad stimulation (right panel), RBC velocity increased. **c** Time series of RBC velocities and flux during whisker pad stimulation. RBC velocity at the precapillary sphincter was significantly higher than the bulb and first order capillary at baseline and peaked around 10 s after stimulation before returning to baseline. **d** Summary of the difference between maximal and baseline RBC velocity during whisker stimulation. In **d**, the LME analysis was performed on log-transformed data to ensure homoscedasticity. $n = 6$ mice, ±SEM, significance code $*p < 0.05$. Source data are provided as a Source Data file.

sphincter than the first order capillary (Fig. 4d). RBC flux through the precapillary sphincter complex increased 25% from baseline to peak stimulation (mean flux increased from $543 \pm 25$ to $679 \pm 50$ cells/s, Fig. 4c). The sphincter, however, retained a pressure-reducing effect during peak stimulation (three-times greater pressure drop per unit length compared to the first order capillary). RBC velocity and flux returned to baseline 20–30 s after ending stimulation (Fig. 4c), concurrent with the post-stimulus undershoot (Fig. 3c, f). Before, during, and after whisker stimulation, we observed passage of single RBCs through the precapillary sphincter, which may optimize hematocrit along the PA[24] and the oxygen delivery to brain tissue (Supplementary Movie 5). Collectively, our data suggest that the sphincter complex augments the reduction of blood pressure from the proximal PAs to downstream capillaries, actively regulates local diameter and RBC flux during functional stimulation, and equalizes the distribution of RBCs entering the upper and lower cortical layers.

**Structural elements support bottleneck function.** The presence of a contractile sphincter-encircling cell supports active tone regulation. However, indentation of the sphincter may also

be supported by passive elements to optimize the force–length relationship[25]. Therefore, we investigated whether passive structural elements constrain dilation at the sphincter by injecting papaverine (10 mM), a strong vasodilator, close to the sphincter (Fig. 5a–c). Papaverine blocks the contractility of mural cells by inhibiting vascular phosphodiesterases[26] and calcium channels[27]. Under these conditions, passive structural elements of the vessel become the main factors that stabilize the vessel wall. Both before and after papaverine injection, the lumen diameter of the sphincter was significantly smaller than that of the bulb and first order capillary (Fig. 5c). Yet, the sphincter demonstrated significantly larger dilation in absolute and relative terms compared to the first order capillary. Structural evidence of passive connective tissue was established by staining coronal slices of NG2-dsRed mice with either a collagen α1 type I (COL1A1) or type IV antibody or Alexa633 hydrazide[28], a marker of elastin (Fig. 5 and Supplementary Fig. 7). Elastin was observed in the tunica intima of PAs and at the precapillary sphincter, but not in capillaries (Fig. 5d). Collagen α1 type-I and type-IV staining was observed in the tunica externa of arterioles, precapillary sphincters, capillaries (Fig. 5e), and venules. Thus, common structural proteins ensheathed the precapillary sphincter. The data indicate that the

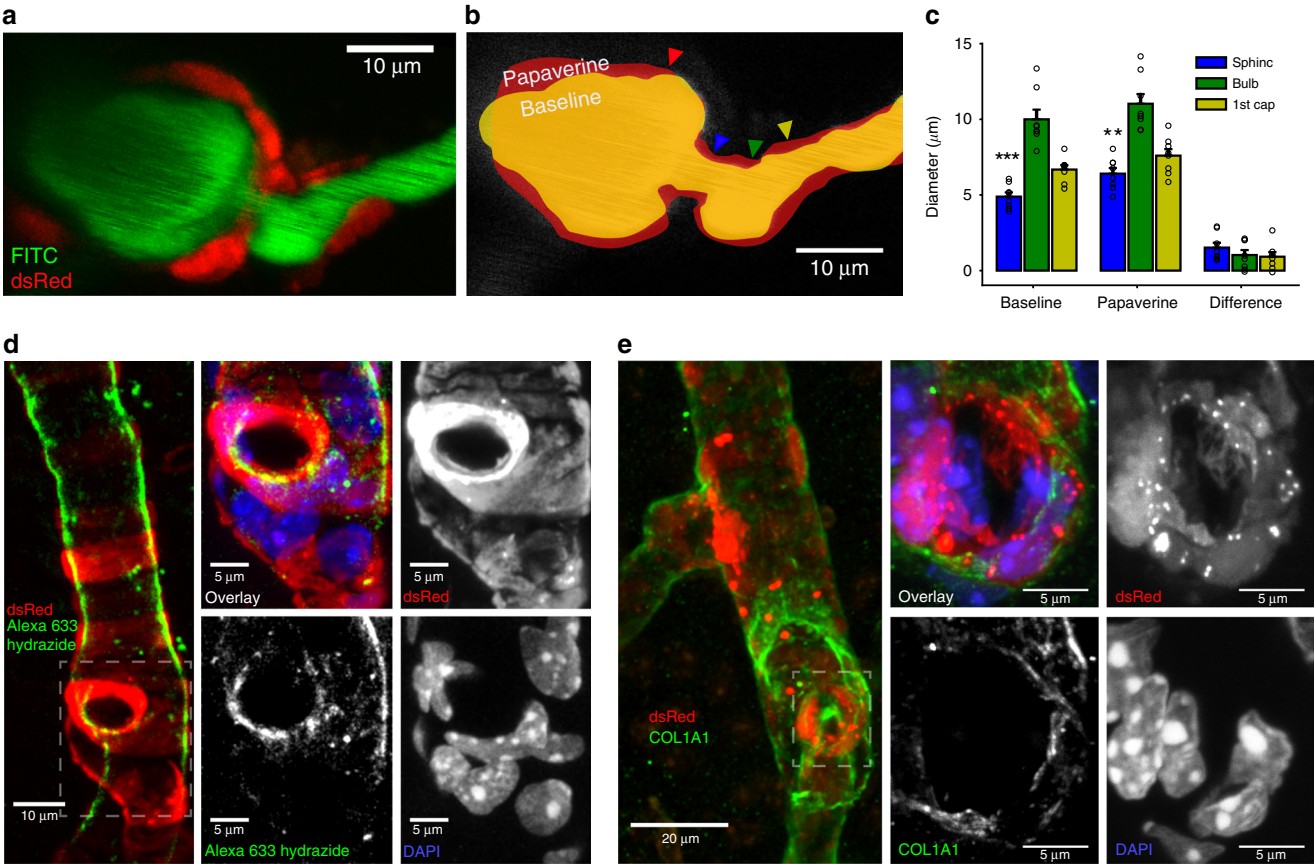

**Fig. 5 Passive structural elements limit vasodilation. a–d** Papaverine (10 mM) was locally injected into the vicinity of precapillary sphincters to dilate the nearby vasculature. **a** Representative maximal intensity projection of an NG2-dsRed mouse PA branch point. **b** Schematic of the papaverine-induced dilation (red) below an outline of the vessel lumen at baseline (yellow). The ROI locations in individual vessel segments are marked by colored arrows. **c** Absolute diameters of vessel segments at baseline and after papaverine addition, and the difference before and after papaverine addition. The baseline dataset was analyzed by the Kruskal–Wallis test, followed by a Wilcoxon rank-sum test (with Holm's p value adjustment) for pairwise comparisons, $n = 8$ mice, ±SEM. The papaverine and difference datasets were analyzed using LME models followed by Tukey post hoc tests for pairwise comparisons. Significance codes **$p < 0.01$, and ***$p < 0.001$. **d** Maximal intensity projections of coronal slices from NG2-dsRed mice stained with Alexa633 hydrazide and DAPI. Left panel: ×20 magnification of a penetrating arteriole with a precapillary sphincter at the branch point. Right panels: ×63 magnification of the precapillary sphincter and first order capillary. Alexa633 hydrazide staining is strong at the sphincter but absent in the first order capillary. **e** Maximal intensity projections of coronal slices of NG2-dsRed mice stained with COL1A1 antibody and DAPI. Left panel: ×20 magnification of a penetrating arteriole with two branches. Right panels: ×63 magnification of the precapillary sphincter at the lower branch. Source data are provided as a Source Data file.

active sphincter is supported by passive structural elements that maintains the lumen indentation and thereby assists in blood pressure reduction from the larger PAs to downstream capillaries both at rest and during stimulation.

**Sphincters constrict in cortical spreading depression.** In the healthy mice considered thus far, precapillary sphincter complexes displayed an active role in blood flow regulation and localized predominantly to the proximal bifurcations of larger PAs (Figs. 3 and 4). As observed for the undershoot (Fig. 3f, i), the flow resistance of the sphincter may greatly increase under physiological conditions (Fig. 2g). This observation prompted the question of whether sphincters constrict in brain pathology. Therefore, we investigated sphincter dynamics during CSD waves that are caused by disrupted brain ion homeostasis and known to cause prolonged vasoconstriction and the migraine aura[7]. Microinjection of 0.5 M potassium acetate in the posterior part of the somatosensory cortex elicited CSD that triggered a triphasic sequence of changes in the diameter and flow of cortical blood vessels: (I) an initial constriction followed by (II) a longer-lasting dilation and (III) prolonged vasoconstriction (Fig. 6a–c, Supplementary Movie 6). The maximal constriction relative to baseline

in phase I was similar among vessel segments, whereas the maximal relative dilation of the precapillary sphincter in phase II was greater than for the bulb and first order capillary (Fig. 6d), but not different from maximal dilation during whisker pad stimulation or local injection of papaverine ($p = 0.71$ Kruskal–Wallis test). During phase III, the precapillary sphincter constricted more (26.2%) than the PA and bulb and doubled flow resistance (Fig. 6e). This was occasionally accompanied by stalling of RBCs at the sphincter, occluding the first order capillary[29] (Supplementary Movie 4), consistent with the increase in flow resistance. The sphincter constriction is likely to be important for the associated long-lasting decreases in cortical blood flow that follow CSD[30]. The structural vulnerability of the sphincter bottleneck during pathological conditions that promote widespread constriction was also observed during cardiac arrest where the sphincter collapsed after ~14–20 min (see Supplementary Fig. 6 and Supplementary Movie 7).

**Sphincters protect capillaries against high pressure.** The blood pressure profile along the microvasculature is practically impossible to measure. However, we reassessed our conclusions about the sphincter properties in a quantitative framework by

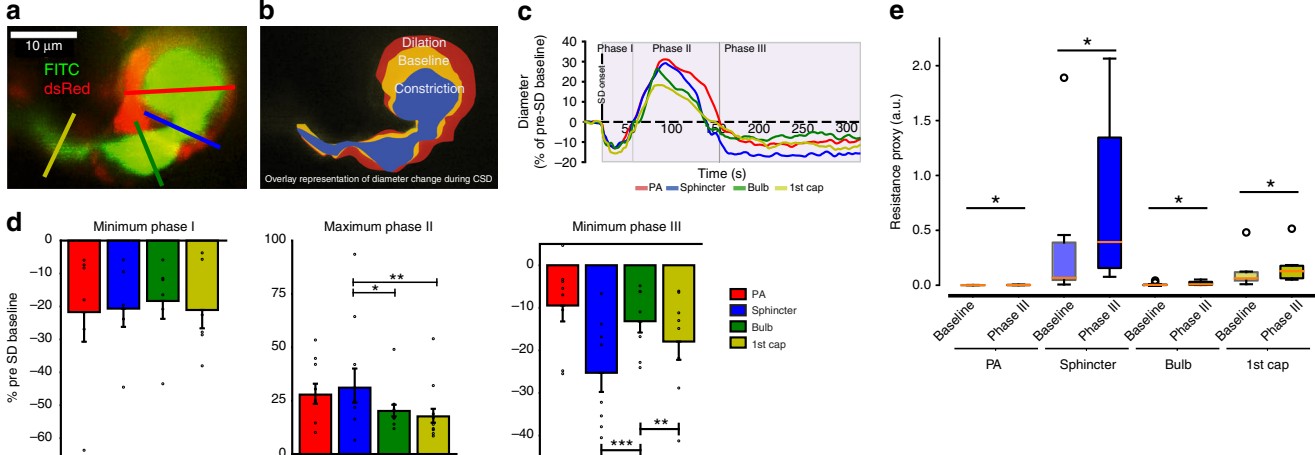

**Fig. 6 Sphincters are vulnerable to cortical spreading depolarization.** Cortical spreading depolarization was elicited in the posterior part of the somatosensory cortex by microinjection of potassium acetate during imaging of the precapillary sphincter. **a** Representative maximal intensity projection of an FITC-dextran loaded NG2-dsRed mouse at a precapillary sphincter. Colored lines mark the ROIs for diameter measures. **b** Overlaid outlines of baseline (yellow), phase II dilation (red), and phase III constriction (blue). **c** Representative time series of diameter changes within vessel segments during the three phases of CSD. **d** Summaries of maximal diameter changes within vessel segments during phase I–III of the CSD. During phase II, the PA and sphincter dilated significantly more than the first order capillary. During phase III, the sphincter constricted significantly more than the PA and the bulb. Datasets were analyzed via LME models, followed by Tukey post hoc tests for pairwise comparisons (phase II data were log-transformed to ensure homoscedasticity). **e** Boxplot summary of the estimated flow resistances at vessel segments at baseline and during phase III of CSD. Paired Wilcoxon signed rank tests were used to establish the difference ($p < 0.05$) before and during CSD phase III. $n = 6$ mice for phase I, 9 mice for phase II and 8 mice for phase III, ±SEM. The box extends from the lower to upper quartile values of the data, with a line at the median. The whiskers extend from the box to show the range of the data. Flier points are those past the end of the whiskers. In each figure, significance codes *$p < 0.05$, **$p < 0.01$, and ***$p < 0.001$. Source data are provided as a Source Data file.

developing a simple blood flow and pressure model of a cortical network with sphincters on the two topmost branches based on image reconstruction of a single PA and its associated branchings into first and second order capillaries (Fig. 7a). The dependency of blood viscosity on diameter and hematocrit was based on prior models[31] (see Methods). Despite the well-known limitations caused by boundary conditions[32], the large pressure reducing effect of the sphincter on blood pressure in the proximal first order capillaries is evident from simulations (Fig. 6a, right; the effect of placing a sphincter in a distal branch was small, i.e., sphincters are only necessary with a large pressure difference between the PA and the first order capillary). The size of the pressure drop across the sphincter was inversely proportional to the diameter of the sphincter (Fig. 7b) under resting conditions and was insignificant without the indentation, i.e., when the diameter at the sphincter was equal to the first order capillary. Applying the relative diameter changes during functional stimulation reduced, but did not eliminate, the pressure drop across the sphincter, in line with our calculations above, using RBC velocity and diameter changes during peak stimulation (Fig. 4c, d). The pressure increase in the PA during stimulation is unknown. Nonetheless, a pressure increase augments the relative increase in pressure drop across the sphincter (compare dark with light green curves in Fig. 7b, right). On an absolute scale, an increasing inlet pressure of the PA also increases the pressure in the first order capillary (Fig. 7c, solid curves). With increasing pressure in the PA (blue to red curves), the sphincter therefore has to contract in order to maintain a low pressure, e.g., below 20 mmHg, in the capillary. This property matches our observation that sphincters are mostly found on proximal branch points of larger PAs, which are expected to carry a higher pressure on average (Fig. 2). The bottleneck function of the sphincter reduces blood flow into the downstream capillaries. Low-flow bifurcations in the microcirculation typically receive less RBCs due to plasma skimming[33]. In accordance, we found that the presence of

sphincters both reduced bulk flow (blue curve) and hematocrit (green curve) into the downstream capillaries (Fig. 2d) using an empirical law of blood phase separation[24].

## Discussion

The organization of the cortical vasculature simultaneously accommodates sufficient pressure for perfusion of each cortical layer and prevents the blood pressure head from inducing tissue damage. Here, we show that precapillary sphincters represent active bottlenecks with high flow resistance, and that they are strategically located at proximal branches of large PAs descending to large first order capillaries in upper cortical layers where microvessels withstand high arterial pressures (Figs. 2 and 6). This localization at just a subset of proximal bifurcations contributes to equalize perfusion to capillary beds along the entire length of the PA by increasing flow resistance into proximal branches as well as increasing plasma skimming. In addition, the reduction of transmural pressure in capillaries downstream from the sphincter protects capillaries and brain tissue against hemorrhage under baseline conditions and during functional activation (Figs. 2–4, 6). The bulb had low pericyte coverage and remained less vasoactive than the precapillary sphincter and first order capillary (Fig. 1a, Supplementary Fig. 4, and Supplementary Movie 2). Yet, the large cross-sectional area of the bulb caused deceleration, deformation, and realignment of RBCs[34] as they entered the capillary network (Supplementary Movie 5). The sphincter location is consistent with the assumption that vascular resistance is higher in the superficial cortical layers and declines over the depth of the cortex[32]. However, the high sensitivity of flow resistance to constriction becomes precarious in pathological conditions that promote general constriction (Fig. 6 and Supplementary Fig. 6).

In principle, the bottleneck structure of the precapillary sphincter can arise from both active contractile elements and passive structural elements. The α-SMA protein is key for

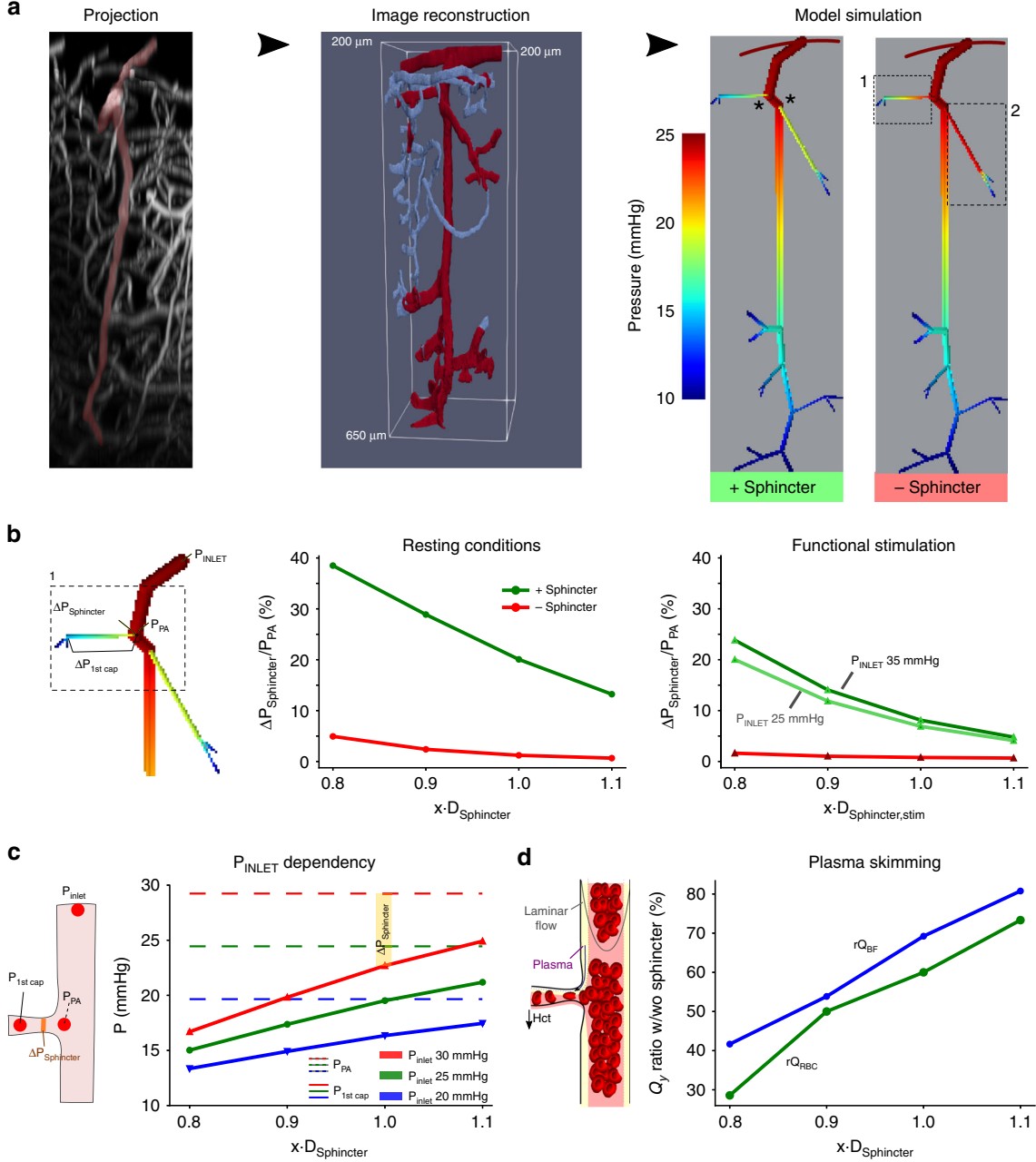

**Fig. 7 Sphincters reduce pressure, flow, and hematocrit into capillaries. a** An xz-projection of a z-stack covering an entire PA (left) was reconstructed (middle) and converted into a computational model of a PA and associated first and second order capillaries. Model simulation (right) including the sphincters (green, marked by asterisks), or without the sphincters (red), observed in the two proximal branches along the PA gave rise to higly divergent pressure profiles along the first order capillaries (highlighted boxes in the model without sphincters). **b** Pressure drop across the sphincter depends on its diameter. Focusing on the first sphincter (left panel), the pressure drop across the sphincter ($\Delta P_{Sphincter}$) relative to the pressure at the branch point ($P_{PA}$) was calculated as a function of sphincter diameters ($x \cdot D_{Sphincter}$) under resting conditions (middle) or upon functional stimulation (right, using the relative changes in dilation from Fig. 3) with the sphincter (green curves) or without (red curves, where diameter of the sphincter is equal to the first order capillary). **c** The degree of sphincter contraction correlates to the pressure in the PA. With increasing inlet pressures into the PA (20 mmHg: blue, 25 mmHg: green, and 30 mmHg: red), the PA pressure increases (dashed curves) and the sphincter must contract to maintain a relatively low pressure into the first order capillary (full curves). The difference between the pressures in the PA and the first order capillary is the pressure drop across the sphincter ($\Delta P_{Sphincter}$, see inset). **d** Flow reduction and phase separation effects due to the sphincter. The ratios of blood flow ($rQ_{BF}$, blue curve) and RBC flow ($rQ_{RBC}$, green curve) with and without the sphincter ($rQ_{with\_Sphincter}/rQ_{without\_Sphincter}$) was calculated as a function of sphincter diameter. Both flows correlate proportionally with diameter but the RBC flow remains lower due to plasma skimming. Source data are provided as a Source Data file.

contractile function, is widely expressed in VSMCs, and is consistently identified in pericytes of first order capillaries within the cortex[18,19,35]. In accordance with previous reports[19], we observed α-SMA along the PA and in some cases up until fourth order capillaries, and within the mural cell encircling the sphincter

(Fig. 3a). In addition, currently available biomarkers of pericytes were unable to identify the sphincter cell as either a pericyte or a VSMC (Supplementary Fig. 7).

While we cannot rule out passive contributions to the sphincter vasoactivity from the vasomotor responses of the adjacent PA, the

presence of α-SMA supports the capacity for active vasomotor responses at the sphincter (Fig. 3c–i). The integrity and morphology of the sphincter was preserved after local administration of papaverine despite significantly greater dilation of the sphincter compared to the bulb and first order capillary (Fig. 5c). The passive and active characterization demonstrates that the sphincter is functionally different from the rest of the first order capillary. The elastin[28] and filamentous collagen α1 type 1 (Fig. 5e, f) expression provide a structural scaffold that optimizes the force–length relationship of the sphincter cell and may support the structural integrity of the sphincter during increases in blood pressure (Fig. 5d, e). The preferential occurrence of sphincters at proximal PA branches suggests that the local angioarchitecture determines the overall distribution of cerebral blood flow between arterioles and capillaries (Fig. 2). The sphincter provides a heamodynamic division between capillary and arterial blood flow that is consistent with the idea that cortical flow control is regulated both in capillaries and arterioles and that regulation of capillary blood flow can occur independently from the arteriolar flow[5,36,37]. However, the sphincter capacity for pronounced diameter changes during functional stimulation allows for considerable dynamical shifts in the distribution of flow resistance[38–40] (Fig. 3), which may reconcile some of the controversies regarding the dynamic regulation of cerebrovascular resistance as described previously[5,35,41,42]. Furthermore, as the sphincter reduces blood flow into the downstream capillaries, the sphincter also increase the relative extent of plasma skimming[24,31], i.e., reduces the hematocrit into the capillaries, that in turn supports redistribution of hematocrit within the local cortical vasculature (Fig. 7d). This redistribution of hematocrit is maintained during functional sphincter dilation.

CSD is a slow depolarizing wave that is involved in migraine, traumatic brain injury, and stroke[43]. CSD evokes an initial vasoconstriction (phase I), immediately followed by a transient hyperemic response (phase II), which is superseded by a long-lasting vasoconstriction of arterioles and capillaries (phase III) that impairs the NVC[7,44]. During CSD, the sphincter exhibited pronounced diameter changes (Fig. 6) and constricted persistently during phase III (Supplementary Movie 6). Persistent sphincter constriction reduced both RBC flux and the hematocrit of the capillary bed. The long-lasting oligaemia previously described in CSD could arise from the high resistance observed at precapillary sphincters[7], and further pharmacological research on this structure could improve the outcome of CSD in the ischemic brain or in patients with migraine.

Precapillary sphincters represent important anatomical sites of blood flow regulation due to their strategic placement at branch points of proximal PAs, where they reduce both pressure and RBC flux into the downstream capillary bed and thereby regulate perfusion along the PA (Fig. 7). The unique location endows larger capacities of control than is achieveable by downstream contractile capillary pericytes. Precapillary sphincters are therefore dissimilar from vascular sphincters[45] that exist along the capillary and at capillary branchpoints, and have confusingly been named either precapillary smooth muscle[46] or contractile capillary pericytes[5,42]. While maximal dilation of the sphincter cell is structurally limited, we show a high capacity for vasomotor control around a baseline diameter of 3–4 μm, where flow resistance is most sensitive to diameter changes. Therefore, precapillary sphincters represent a mechanism to equalize pressure and RBC flux between the capillary networks that branch off from the upper, middle, and lower parts of the PA. Simultaneously, sphincters protect downstream capillaries and brain tissue against adverse blood pressure. During pathology, sphincter constriction limited perfusion of downstream capillaries. Prevention of sphincter constriction may be of therapeutic importance in migraine, cerebral ischemia, and dementia[47].

## Methods

**Animal handling.** Animal procedures were approved by The Danish National Ethics committee according to the guidelines set forth in the European Council's Convention for the Protection of Vertebrate Animals Used for Experimental and Other Scientific Purposes, and we have complied with all relevant ethical regulations for animal testing and research. A total of 38 male or female NG2-dsRed mice (Tg(Cspg4-DsRed.T1)1Akik/J; Jackson Laboratory; 19–60 weeks old) and 27 male or female wild-type mice (C57bl/6j; Janvier-labs, France; 16–32 week) were used. The NG2-DsRed mice were used in the whisker pad stimulation, cardiac arrest, thinned skull, and local ejection of papaverine studies. The rest of the studies were performed in wild-type mice.

**Surgical procedures.** Anesthesia was induced with intraperitoneal (i.p.) bolus injections of xylazine (10 mg/kg) and ketamine (60 mg/kg) and maintained during surgery with supplemental doses of ketamine (30 mg/kg/20 min, i.p.). Mechanical ventilation (Harvard Apparatus, Minivent type 845) was controlled through a cannulation of the trachea. One catheter was inserted into the left femoral artery to monitor blood pressure and to collect blood samples. Another catheter was inserted into the femoral vein to administer chemical compounds. The content of blood gasses in arterial blood samples (50 μL) was analyzed by an ABL700 (Radiometer, Copenhagen; $pO_2$, normal range: 95–110 mmHg; $pCO_2$, normal range: 35–40 mmHg; pH, normal range: 7.35–7.45). To maintain physiological conditions, both respiration and the mixed air supply were adjusted according to the blood gas analysis, or occasionally according to continuously monitored end-expiratory $CO_2$ (Harvard Apparatus, Capnograph 340) and blood oxygen saturation (Kent Scientific, MouseStat pulsoximeter). A craniotomy (diameter ~3 mm. Center coordinates: 3 mm right of and 0.5 mm behind bregma) was drilled above the right somatosensory barrel cortex. We switched the anesthesia to α-chloralose (33% w/vol; 0.01 mL/10 g/h) upon the completion of surgery. At the end of the experiments, mice were euthanized by intravenous injection of pentobarbital followed by cervical dislocation.

To ensure that the precapillary sphincters were not a result of the craniotomy, we made thinned skull preparations over the barrel cortex at the point of the surgical procedure where we would otherwise have made a craniotomy. We thinned the skull to approximately 40 μm thickness, polished it with tin oxide powder, and covered the window with agarose and a coverslip.

**Chronic cranial window implantation.** A chronic cranial window was installed approximately 3 weeks prior to imaging in mice with a C57Bl/6 background. The surgical procedure was adapted from Goldey et al.[48]. A small craniotomy was performed over the left barrel cortex under isoflurane anesthesia and a custom-made reinforced cover glass consisting of three 3 mm coverslips glued on top of each other and onto a 5 mm coverslip was installed. A custom-made head bar was attached to the right side of the skull, allowing for head immobilization during imaging sessions. In the 5 days following implantation, the animal was closely monitored and treated for pain and infection as described of Goldey et al.[48]. Training for imaging experiments could commence after the animal had recovered after surgery. The animal was familiarized with the experimenter through gentle handling. After several handling sessions, and when the animal was comfortable with the experimenter, it was slowly accustomed to head immobilization. The animal was given treats in the form of sweetened condensed milk during the training process. After the animal had been habituated with the head immobilization for periods of about an hour in length, they were ready for imaging experiments.

**Electrical stimulation of the whisker pad.** The mouse sensory barrel cortex was activated by whisker pad stimulation. The contralateral ramus infraorbitalis (IO) of the trigeminal nerve was stimulated electrically using a set of custom-made bipolar electrodes inserted percutaneously. The cathode was positioned relative to the hiatus IO, and the anode was inserted into the masticatory muscles. Thalamo-cortical IO stimulation was performed at an intensity of 1.5 mA (ISO-flex; A.M.P.I.) for 1 ms in trains of 20 s at 2 Hz.

**Pressure ejection of papaverine via glass micro-pipette.** Borosilicate glass micro-pipettes were produced by a pipette puller (P-97, Sutter Instrument) with a resistance of 2.5–3.0 MΩ. The pipette was loaded with a mixture of 10 μm Alexa 594 and 10 mM papaverine in order to visualize the pipette tip using both an epi-fluorescent camera and two-photon microscope. Guided by two-photon microscopy and operated by a micromanipulator, the pipette was carefully inserted into the cortex to minimize tissue damage and avoid vessel bleeding. The distance between the pipette tip and vasculature was 30–50 μm. Papaverine was locally ejected for ~1 s three times using <15 psi air pressure in the pipette (PV830 Pneumatic PicoPump, World Precision Instruments). A red cloud (Alexa 594) ejected from the pipette tip was visually observed to cover the local vascular region simultaneously, and the background returned to normal approximately 1 min after

puffing[6]. Papaverine was preconditioned for 5 min before imaging the same vasculature again.

**Cortical spreading depression**. In a subset of experiments, CSD was triggered 2 mm away from the recording site using pressure injection of 0.5 M potassium acetate (KAc) into the cortex (estimated volume ~0.5 μL). Apart from triggering CSD, KAc injection did not cause a brain lesion (bleeding or tissue damage). In addition, our technique for making craniotomies was validated previously by measuring cerebral blood flow while drilling the craniotomy, ensuring that no spreading depolarization was triggered during surgical preparation. Using this technique, we could show that no spreading depolarization was elicited[44]. Moreover, prior to our first spreading depolarization, we measured whisker responses, and mice that did not exhibit normal vessel diameter dilation, i.e., less than 5% dilation, to whisker stimulation were discarded from the dataset, ensuring that no spreading depolarization were triggered before recordings started.

**Two-photon imaging**. Images and videos were obtained using two sets of laser-scanning two-photon microscopes. Experiments to quantify prevalence, measure RBC velocity, CSD, and cardiac arrest were performed using a commercial two-photon microscope (FluoView FVMPE-RS, Olympus) equipped with a 25 × 1.05 NA water-immersion objective (Olympus) and a Mai Tai HP Ti:Sapphire laser (Millennia Pro, Spectra Physics). Whisker pad stimulation and papaverine ejection experiments were performed using a second commercial two-photon microscope (Femto3D-RC, Femtonics Ltd.) with a 25 × 1.0 NA water-immersion objective with piezo motor and a Ti:Sapphire laser (Mai Tai HP Deep See, Spectra-Physics). The excitation wavelength was set to 900 nm. The emitted light was filtered to collect red (590–650 nm) and green (510–560 nm) light from dsRed (VSMC/pericytes) or SR101 (astrocytes) and FITC-dextran (vessel lumen), respectively.

The prevalence of precapillary sphincters and bulbs was studied by acquiring image stacks using our Olympus two-photon microscope (Fluoview) and tracking each PA from pial to >650 μm in depth in frame-scan mode at around 1 frame per second with a pixel resolution of 512 × 512 at an excitation wavelength of 1000 nm. RBC velocity was measured in resonance bidirectional line-scan mode with a scan rate of 15,873 Hz (0.063 ms per line) and pixel resolution of 512 pixels per line. CSD was imaged in one vessel branching from the PA to a first order capillary, including the neck and bulb structure in a single plane. The excitation wavelength was set to 920 nm, the frame resolution was 0.255 μm/pixel with a 512 × 384 pixel frame, and images were taken at a speed of 0.81 frames per second for CSD.

In the experiments using the Femtonics microscope, we recorded the whole volume including the vessel segments of interest by fast repetitive hyperstack imaging (4D imaging) with continuous cycles of image stacks along the z-axis. This compensates for focus drift and studies vasculatures spanning a certain z-axis range. Each image stack was acquired within 1 s and comprised 9–10 planes with a plane distance of 3–4 μm. This covered the whole z-axis range of the investigated blood vessels. The pixel sizes in the x–y plane were 0.2–0.3 μm.

**Two-photon imaging analysis**. Data were analyzed in ImageJ or MATLAB using custom-built software. RBC velocity was determined using the velocity_from_tif.m MATLAB script[49]. To determine the prevalence of precapillary sphincters and in the cardiac arrest experiments, multiple ROIs were manually placed across the vessel lumen in ImageJ, measuring vessel diameters. In the CSD experiments, rectangular ROIs with a width of 2 or 4 μm were drawn perpendicular to the surface of the vessel at the defined locations. An active contour algorithm (Chan-Vese segmentation) was used to calculate the change in vessel diameter in these ROIs. The diameter change over time was detected for each ROI. For the 4D imaging performed in the whisker pad stimulation and papaverine ejection experiments, each image stack was flattened onto one image by maximal intensity projection, which converts the data to the same formats of CSD. Similar diameter analysis methods were used. Values from each ROI type were averaged per mouse. Arteriole bifurcations leading to two equally sized arterioles and first order capillaries bifurcating <10 μm from the arteriole branch point were not included in the analysis. 3D renderings for Supplementary movies were created with Amira software (ThermoFisher Scientific).

To estimate the change in flow resistance upon changes in vessel diameter, we combined Poiseuille's law ($\Delta P = \frac{8\mu L Q}{\pi r^4}$, where $P$ is pressure, $\mu$ is dynamic viscosity, $L$ is length, $Q$ is flow, and $r$ is vessel radius) and the standard hemodynamics measure of flow resistance ($R$) in laminar fluid flow ($\Delta P = RQ$). We assume that the specific dynamic viscosity is the same between the PA and the Sphincter, Bulb, and first order capillary. A resistance proxy per unit length ($R_{proxy}$) therefore becomes $R_{proxy} = \frac{1}{r^4}$.

**Hemodynamic network modeling**. We developed a simple hemodynamic network model based on reconstruction of a PA and associated first and second order capillaries. The VMTK toolbox (version 1.4.0) was used for reconstruction from which the nodes between vessel segments (network edges) were extracted. The diameter and length of each meandering vessel segment was estimated from image measures of diameters. In a network, Kirchoff's law states that the sum of flows entering and leaving any internal node equals zero, $\sum_j Q_j = \sum_j \frac{\Delta P_j^n}{R_j^n} = 0$, where $Q_j^n$

is the flow, $R_j^n$ is the vascular flow resistance, and $\Delta P_j^n$ is the pressure drop in the $j$th vessel entering the $n$th node. Assuming that the vessels are rigid and the flow laminar, the flow resistances were calculated using Poiseuille's law (see above). We applied the empirical model describing the changes in apparent viscosity of blood ($\mu$) with diameter ($D$), and discharge hematocrit ($H_D$)[50,51]

$$C = \left(0.8 + e^{(-0.075D)}\right) \times \left(\frac{1}{1 + 10^{-11} \times D^{12}} - 1\right) + \frac{1}{1 + 10^{-11} \times D^{12}}, \quad (1)$$

$$\mu_{0.45} = 220 \times e^{(-1.3D)} + 3.2 - 2.44e^{(-0.06D^{0.645})}, \quad (2)$$

$$\mu = 1 + (\mu_{0.45} - 1)\frac{(1 - H_D)^C - 1}{(1 - 0.45)^C - 1}. \quad (3)$$

$H_D$ was calculated based on a tube hematocrit ($H_T$) of 0.3[31,52]

$$\frac{H_T}{H_D} = H_D + (1 - H_D)\left(1 + 1.7e^{(-0.415D)} - 0.6e^{(-0.011D)}\right). \quad (4)$$

To solve the system of linear equations, we chose the boundary conditions such that inlet pressure into the PA was 25 mmHg in the control situation and outlet pressures out of second order capillaries were 10 mmHg. The system was solved using the root solver in SciPy (1.1.0).

*Phase-separation at bifurcations*: At diameters below ~30 μm, the distribution of RBCs at microvascular bifurcations does not follow the overall blood flow distribution. Low-flow bifurcations typically receive reduced hematocrit, i.e. plasma skimming, which has been empirically described in vivo[24,31]

$$A = -13.29\frac{(D_A^2/D_B^2 - 1)}{(D_A^2/D_B^2 + 1) \times D_P} \times (1 - H_D), \quad (5)$$

$$B = 1 + 6.98 \times (1 - H_D)/D_P \quad (6)$$

$$\frac{Q_{RBC,A}}{Q_{RBC,P}} = \frac{1}{1 + e^{-(A + B \cdot \text{logit}(Q_A/Q_P))}}, \quad (7)$$

where $D_A$, $D_B$, and $D_P$ are the diameters of the two daughter vessels and parent vessel, respectively. $Q_A$, $Q_p$ are blood flows of the daughter and parent vessel, respectively $\frac{Q_{RBC,A}}{Q_{RBC,P}}$ is the fraction of RBC flow into the daugther vessel and logit($x$) $= 1/(1 - x)$.

*Limitations*: All assumptions underlying Poiseuille's law apply. The boundary conditions have a strong influence on the solution as the system is forced to comply with the preset boundary pressures. Hence, the overall pressure distribution along the PA is largely determined from the onset, severely limiting the analysis of pressure along the PA. However, the effect of changing sphincter diameter and/or disposing the sphincter on the pressure in the downstream capillaries is evident. We have assumed that the empirical formulas to calculate viscosity and phase separation applies to the cerebral microcirculation of mice.

**Immunohistochemistry**. Adult NG2-dsRed mice were transcardially perfused with 4% paraformaldehyde (PFA) and their brains extracted and cryoprotected in 30% sucrose, rapidly frozen in cold isopentane (−30 °C), and sectioned into 25 and 50 μm thickness using a cryostat. Sections were rinsed for 5 min three times in 0.1 M phosphate-buffered saline (PBS) and, for collagen-I staining, antigen retrieval performed using hot citrate buffer (90 °C, pH 6.0) for 20 min. The 50-μm sections were permeabilized and blocked in 0.5% Triton-X 100 in 1× PBS (pH 7.2) and 1% bovine serum albumin (BSA) overnight at 4 °C, whereas 25-μm sections were permeabilized in 0.5% Triton-X 100 in 1× PBS for 30 min and blocked in 5% NGS, 5% BSA, and 0.5% Triton-X 100 in 1× PBS for 1 h at room temperature (RT). Sections were incubated for two nights at 4 °C in primary antibodies in blocking buffer containing 1–5% BSA and 5% NGS in 0.25–0.5% Triton-X 100 in 1× PBS. The following primary antibodies were used: mouse ACTA2-FITC (1:200; Sigma; F3777), rabbit anti-collagen-I (1:50; ab34710), rat anti-vitronectin (1:100; #347317; R & D systems), goat anti-aminopeptidase N/CD13 antibody (1:100; AF2335; R & D systems), rat anti-MCAM/CD146 antibody (1:100, R & D Systems, MAB7718), rabbit anti-collagen antibody, Type IV (1:100, Merck Millipore, AB756P). Elastin was labeled using an artery-specific red dye, Alexa Fluor 633 (A30634, Thermo-Fisher Scientific) at 1:300 dilution from 2 mM stock. Alexa Fluor 633 was added to the brain sections for 10 min and then rinsed. The sections were then washed for 5 min three times in 0.1 M PBS and incubated with secondary antibodies: goat anti-rabbit Alexa488 (1:500; Thermo Fisher SCIENTIFIC, TC252465), goat anti-mouse Alexa488 (1:500, ThermoFisher SCIENTIFIC, #1726530), rabbit anti-rat Alexa488 (1:500, ThermoFisher SCIENTIFIC, #1717038) or chicken anti-goat Alexa488 (1:500, ThermoFisher SCIENTIFIC, #1932500) for 1 h at RT. After incubation with secondary antibody, the sections were rinsed for 5 min three times in 1× PBS, incubated in Hoechst (1:6000) for 7 min, rinsed again (3 × 5 min) in 1× PBS, and mounted using SlowFade™ Diamond Antifade Mountant (Invitrogen; S36963). Fluorescence images were acquired with a confocal laser scanning microscope (LSM 700 or 710) equipped with Zen software and ×20/0.8 NA and ×63/1.40 NA oil DIC M27 objectives at ×1 (0.170 μm/pixel) and ×4 (0.021 μm/pixel) digital zoom, respectively. Care was taken to ensure similar fluorescence across images. The Nissl neurotrace 500/525 (1:25, ThermoFisher SCIENTIFIC) marker of

fusiform pericytes[20] was used in in vivo two-photon imaging. Prior to imaging, Nissl neurotrace 500/525 was loaded topically for 5 minutes, washed out thoroughly and imaged 1–4 h later after i.v. injection of cascade blue (ThermoFisher SCIENTIFIC).

**Statistical analysis**. Datasets are presented as mean ± SEM , standard box plots, or in the case of log-transformed data as back-transformed means ± 95% confidence intervals. The normality of data was assessed using Shapiro–Wilk and graphical tests. For normal datasets, linear mixed effects (LME) model analyses were performed. LME was chosen to take proper advantage of multiple measurements of parameters and/or multiple time points in the same animal. Vessel segments (PA, sphincter, bulb, and first order capillary) were included as the fixed effect, whereas the particular mouse and vessel branch were included as random effects as needed. Heteroscedastic datasets were log-transformed to conform to analyses as indicated. Significant differences ($p$ value < 0.05) were obtained by likelihood ratio tests of the LME model with the fixed effect in question against a model without the fixed effect. Tukey's post hoc test was used for pairwise comparisons between elements in the fixed effect group. For non-normal data, nonparametric Wilcoxon signed-rank tests were used for paired samples, whereas the Kruskal–Wallis test was used for multiple independent groups. For pairwise comparisons, the Wilcoxon rank-sum test with the Holm's $p$ value adjustment method was used. Finally, linear regression was used to assess the relationships and fitted to datasets. All statistical analyses were performed using R (version 3.4.4; packages lme4[53] and dplyr) and Prism version 5.

**Reporting summary**. Further information on research design is available in the Nature Research Reporting Summary linked to this article.

## Data availability

The data that supports the findings of this study are available from the corresponding author upon request. The source data underlying Figs. 2b–f, 3d–i, 4c, d, 5c, 6d, e, 7b–d, Supplementary Table 1 and Supplementary Figs. 5, c–f and 6c, d are provided as a Source Data file.

## Code availability

The custom made code used for data analysis is available from the corresponding author upon request.

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

## Acknowledgements
We would like to acknowledge our animal technician Micael Lønstrup for his help with animal experiments. Nikolay Kutuzov and Dr. Krzysztof Kucharz for scientific discussions. Thanks to Dr. Kirsten Thomsen for advice regarding the statistical analysis. A special thanks to Prof. Anna Devor for hosting and supervising Aske Graakjær Krogsgaard Jensen during his visit in San Diego working with awake mice. Thanks to the core facility for integrated microscopy (CFIM) at our institution for their service. This study was supported by the Lundbeck Foundation, the Danish Medical Research Council, the Alice Brenaa Fondation, and a Nordea Foundation Grant to the Center for Healthy Aging.

## Author contributions
All authors contributed to designing the study, doing the experiments, analyzing the results, and writing the paper.

## Competing interest
The authors declare no competing interests.
