## [Peer Review File · Nature Communications]

Editorial Note: Parts of this Peer Review File have been redacted as indicated to remove third-party material where no permission to publish could be obtained

Reviewers' comments:

Reviewer #1 (Remarks to the Author):

The present paper, using *in vivo* 2-photon microscopy in mice expressing NG2 as a marker of vascular mural cells, identifies precapillary structures, akin to those described by scanning electron microscopy and microvascular casts (see below), consisting of the terminal part of a penetrating arteriole, a short constricted segment (sphincter) followed by a dilatation (bulb) merging into a first order capillary. *In vivo* imaging and immunocytochemistry revealed expression of NG2, alphaSMA, lectin, COL1A1, and elastin. Electrical stimulation of the whisker pad or local injection of the vasorelaxant papaverin led to a dilatation of the sphincter. Calculation of RBC velocity and flux suggested that the sphincters are sites of high vascular resistance with the potential of regulating capillary flow. The sphincters constricted, relaxed and constricted again after cortical spreading depression, reflecting the temporal pattern of flow changes associated with this condition, and constricted with a delayed time course after cardiac arrest. Grubb et al. conclude that sphincters are bottlenecks for capillary flow.

This is technically-sound and well-written paper that, however, raises concerns about the novelty of some of the observations, and the limited morphological, cellular and mechanistic advances provided by the data.

1. Vascular sphincters have been long described in the cerebral circulation and implicated in the regulation of cerebral blood flow in health and disease (e.g., Fed Proc 34: 1468, 1975; Stroke 1981;12:653; K. Nakai et al. Scannig Microscopy 1989; Anat Rec. 1998;251:87; American Journal of Anatomy.1978;153:523; J Neurosurg. 2013;118:763; a discussion of potential role of sphincters in CSD can be found in: Neuroimage. 2011;56:1001). The functional impact of these structures on capillary perfusion were articulated in the arteriolar "stopcock theory" (Neuroimage. 2011;56:1001). Therefore, the conceptual novelty of the paper is limited.

2. The morphological, cellular and molecular description of these structures is also limited, and several questions remain to be answered:

(a) How frequent are these sphincters? Are they located on terminal segment of all penetrating arterioles and only a fraction of them? This information is critical because it will give a sense of their potential to impact the flow at the regional level and to protect the capillary bed, as suggested in the paper.

(b) Are there actually cells with contractile proteins located at the stricture? If so, what is their molecular signature? NG2 is present near the stricture, but NG2, a proteoglycan, is a non-specific marker which happens to be also in smooth muscle cells and pericytes, with little relevance to the purported function of these structures.

(c) Are the cellular elements in question embedded in the endothelial basement membrane, like pericytes, or are outside of it, like smooth muscle cells? What are their ultrastructural characteristics?

(d) Elastin, COL1A1 and lectins are abundant in the cerebral vasculature. Smooth muscle actin may help with the identification of the cells and suggest contractile function, but the evidence presented is not convincing, the immunolabel seemingly being located also in endothelial cells (fig. 3a).

(e) There have been major advances in the molecular signature of mural cells, and a number of useful markers have been identified (Nature 2018;80:475; Nat Neurosci. 2017;79:559; Sci Rep. 2016;6:35108). This new knowledge could be used to provide a better understanding of the cellular identity and nature of these structures.

3. The data show dilatation of these structures during neural activity and exposure to papaverine. However, these manipulations will also dilate the arteriole in continuity with these structures. Therefore, the observed dilatation and constriction could simply represent the diameter change of the terminal segment of the parent arteriole, and the size modification of the stricture could be

related to mechanical forces transmitted from the arteriolar system upstream. The same could be said of the changes in size observed with CSD or cardiac arrest.

4. The impact of the observations of single capillaries, presented in the paper and in the movies, on the overall tissue perfusion and purported protection of capillaries remains to be established and will depend on how frequent these structures are.

5. Although the data on CSD and cardiac arrest are of potential interest, the findings are descriptive and anticipated based on previous literature, and there are no experiments on the mechanisms regulating the size of the sphincters and on their impact on the pathological changes that could derive from their engagement.

Reviewer #2 (Remarks to the Author):

In this manuscript, Grubb et al. provide novel findings in the cortical microvasculature of the brain that identify "precapillary sphincters" between penetrating arterioles and the 1st order vessel. Using live imaging with 2-photon microscopy and immunostaining on brain slices, the authors provide an elegant and convincing structural and functional characterization of this vascular segment at baseline, during functional hyperemia and pathological conditions. The authors conclude that these sphincters represent a mechanism to equalize perfusion to capillary beds along the length of penetrating arteries (PA) and to protect brain tissue against adverse blood pressure change.

This is a novel and interesting study that may be of interest to others in the community. Overall the data shown are solid. However, there are weaknesses and other issues that need to be addressed.

1) Results shown on figure 2 are unclear and confusing.

a) The terminology "PA number" and "1st order capillary" is confusing. In particular, controversy has surrounded the definition of capillary versus arteriole, and pericytes versus smooth muscle cells.

b) Despite the numerous metrics displayed in this figure, some issues are still unclear. Do these sphincters occur only at the boundary between the PA and its 1st branch. As suggested by the authors, the mouse cortex is supplied by short and long cortical arteries. Does the distribution of sphincters differs between short and long cortical arteries.

c) The imaging depth of 2-PM is usually limited to 0-600 μm in the cortex. Was the analysis of the localization of sphincters within the cortex performed *ex vivo* or *in vivo*?

2) On figure 3, it is unclear how flow resistance in the PA, sphincter, bulb and 1st capillary was estimated. What do the authors mean by "proxy resistance". Based on the measured RBC velocity and the Poiseuille's law, the authors elaborate on the pressure drop in the sphincter and the 1st order capillary. This needs to be further explained, especially during whisker stimulation where both velocity and diameter increase in the sphincter. In general, panel 2g and this paragraph are hard to follow.

3) The authors analyzed vasomotor responses of the sphincters during prolonged cardiac arrest (> 20 minutes), which relevance is highly questionable. Hence, I would recommend to delete this experiment and rephrase the title accordingly. Instead, given the proposed role of these sphincters in the protection from high pressures and the perfect *in vivo* imaging model to analyze this, the vasomotor response of these sphincters to acute blood pressure increase or decrease, ie in the context of cerebral flow autoregulation, must be assessed.

4) A mathematical modeling of pressure changes from the PA to the first order vessel at baseline and upon blood pressure changes would strengthen the conclusion that the sphincter complex protects capillaries from blood pressure peaks.

5) The repeated statements throughout the Results section that precapillary sphincters "contribute to pressure distribution", "play an active role in the protection of downstream capillary" etc..

should be deleted and reserved to the discussion.

Reviewer #3 (Remarks to the Author):

Grubb and colleagues have investigated the role of a specific cell on the vascular tree called the precapillary sphincter and how it could regulate blood flow. They nicely use the NG2-DsRed mouse, vascular tracing and two-photon imaging to visualize the sphincter and provide a clear definition as to what constitutes the sphincter. They show that during whisker stimulation in mice that the sphincter can relax leading to an increase in blood flow to the downstream capillary bed and that surrounding the sphincter were structural elements such as elastin and collagen to protect the brain against pressure-induced injury. The papaverine experiments were technically demanding but well performed. Subsequently, the authors go on to show that the sphincter contracts during both induced spreading depolarisations and following cardiac arrest which would limit the capillary blood flow in these conditions. Overall, this study utilizes an excellent imaging setup to visualize changes in the pre-capillary sphincter during neurovascular coupling and pathology, ultimately influencing blood flow.

The techniques used in this manuscript are sound, highly validated and are suitable for the aims of the manuscript. The conclusions are supported by the evidence, and the use of in vivo techniques were appropriate and strengthened these conclusions. With regards to statistics, most of the statistical analysis appears correct, but no apparent power calculation was performed to determine sample sizes.

The role and even the presence of a precapillary sphincter in the cerebrovasculature is still controversial. This novel manuscript sheds some light on the existence of these structures and their influence on blood flow and also adds to the debate about the role of pericytes vs vascular smooth muscle cells on controlling cerebral blood flow.

Specific comments below:

1. Given the placement of the precapillary sphincter at a branch point of a penetrating arteriole and the first order capillary, they are placed in an ideal position to influence capillary blood flow. However, is the name "precapillary sphincter", a misnomer? Are these in fact just contractile pericytes or ensheathing pericytes as they are now known? What is the difference between the action of the so-called precapillary sphincter to the action of the downstream capillary pericytes which actively dilate the blood vessel upon neuronal stimulation? Some discussion was provided on lines 239-243 about the naming of this. Some greater elaboration and discussion about the NG2+ α -SMA+ cells at the sphincter is required and whether they are just a VSMC or a pericyte or a hybrid cell, and how this specific cell type fits in with the continuum of VSMCs to pericytes.
2. Line 57, "close inspection revealed that PAs were covered with VSMC/pericyte hybrids (Fig 1e)". I agree with the authors in that there is a continuum of mural cell cyto-architecture from pial arterioles to 3rd order capillaries. But I disagree that the mural cells covering PAs are VSMC/pericyte hybrids, they should be referred to as VSMCs as that is what the imaging suggests and the depiction in Fig 1E is misleading with a slight protrusion of cell bodies on the PAs.
3. The definition of the sphincter is crucial for the analysis. It is clearly stated that a sphincter is defined by DsRed-positive cells encircling an indentation of the lumen at PA branch points and usually followed by distention of the lumen (the bulb). However, if there is no bulb is it considered a sphincter? The analyses presented in Fig 2 suggest that <50% of branch points off the PA had a sphincter or a bulb. Then how important is the sphincter to overall changes in flow if the sphincter/bulb is not present at the majority of branch points. At the branch points with no bulb, were there DsRed-positive cells where the sphincter would be? Do these vessels dilate in response to stimulation? Some data here on these vessels without sphincters would be informative and some discussion around why only the minority of branch points would have sphincters is needed.
4. Hall et al 2014 Nature showed in their landmark paper that capillaries dilated before PAs in

- response to whisker stimulation. In terms of onset of dilation post-stimulation, how did the sphincter dilation compare to the timing of PA and 1st order capillary dilation? If it is delayed, could this then just be a passive response to the dilation from the capillary and the PA?
5. Line 161 states that in phase II the sphincter had greater dilation ($39\pm 8\%$) than the bulb and 1st order capillary ($22\pm 3\%$ and $21\pm 4\%$ respectively). However, these data do not match the dilation represented in the graph in Fig 5d (sphincter looks around 31%, bulb looks around 20% and 1st order cap. looks around 18%). Please clarify this inconsistency.
 6. The representative figure in Fig 5c shows oligemia out to 300 seconds after the onset of depolarisations with KAc. How long-lasting is the sphincter constriction during phase III beyond 300 seconds and when does this resolve itself? Are the resistance calculations in Fig 5e determined from the minimum diameter change during phase III, or incorporate a measurement of diameter change over time to calculate a chronic resistance change, as this is not clear.
 7. The methods for the cardiac arrest experiments are not well described. The methods state that mice were euthanized by iv injection of pentobarbital but nothing else was mentioned about imaging beyond then. Fig 6 shows that vascular changes were gradual with a final collapse around 25mins following pentobarbital administration. Do you have heart rate data to show that at what stage of the pento delivery was heart rate stopped, was it immediately? Also, the sphincter data would be related to the massive blood pressure drop from cardiac arrest which affects the PA which goes on to effect the sphincter and downstream capillaries. There is no description of an active sphincter constriction, just a delayed one due to presumably the blood pressure drop. Therefore, I am not really sure what these data are showing about specific sphincter biology relative to the PA and capillaries and I am not sure adds much to the manuscript. However, what would be interesting if this is a reversible effect (think of cardiac arrest patients who are resuscitated) and how the sphincter would react then with a restoration of blood flow? Do they remain constricted as described in the pericyte literature during reperfusion following stroke which would limit blood flow to the capillary bed? This would be of much greater interest to the field.
 8. Line 200-1 states that "we observed α -SMA along the PA and 1st-4th order capillaries. However, Ext data fig 4b second panel doesn't show any α -SMA labelling in 4th order capillary (in fact this capillary based on the scale is around $10\mu\text{m}$ which is very large for a 4th order capillary). The large panel in Ext data fig 4a shows α -SMA labelling only down to the 2nd order. This needs to be corrected in the text and the title of extended data figure 4.
 9. Both male and female mice were used as well as a large age range, could these have influenced outcome?
 10. Line 277, "to ensure that the precapillary sphincters were not a result of the craniotomy". I am not sure how a craniotomy would cause a precapillary sphincter, but it is important to perform some experiments with an intact skull to overcome any effects of a craniotomy.
 11. Line 326, "mice that did not exhibit normal vessel diameter responses to stimulation were discarded". Please define what is normal, is it a certain % increase in diameter? Please add the criterion to the methods.
 12. Is reference 30 the same as reference 18?
 13. In Fig 3, * are used to represent statistical significance but this is not mentioned in the panel or the legend.
 14. Extended data table 1, need to specify the n number of mice and vessels analysed in this table.
 15. Extended data figure 3 shows the awake mouse and the presence of the bulb adjacent to a sphincter. However, this is not a DsRed mouse and so the DsRed cells can not be visualized. It would be nice to have an awake NG2-DsRed mouse imaged to confirm that at each bulb, there is a DsRed+ cell at the adjacent sphincter.

Reviewers' comments:

Reviewer #1 (Remarks to the Author):

Rebuttals to reviewer 1 are written in green

The present paper, using in vivo 2-photon microscopy in mice expressing NG2 as a marker of vascular mural cells, identifies precapillary structures, akin to those described by scanning electron microscopy and microvascular casts (see below), consisting of the terminal part of a penetrating arteriole, a short constricted segment (sphincter) followed by a dilatation (bulb) merging into a first order capillary. In vivo imaging and immunocytochemistry revealed expression of NG2, alphaSMA, lectin, COL1A1, and elastin. Electrical stimulation of the whisker pad or local injection of the vasorelaxant papaverin led to a dilatation of the sphincter. Calculation of RBC velocity and flux suggested that the sphincters are sites of high vascular resistance with the potential of regulating capillary flow. The sphincters constricted, relaxed and constricted again after cortical spreading depression, reflecting the temporal pattern of flow changes associated with this condition, and constricted with a delayed time course after cardiac arrest. Grubb et al. conclude that sphincters are bottlenecks for capillary flow.

This is technically-sound and well-written paper that, however, raises concerns about the novelty of some of the observations, and the limited morphological, cellular and mechanistic advances provided by the data.

1. Vascular sphincters have been long described in the cerebral circulation and implicated in the regulation of cerebral blood flow in health and disease (e.g., Fed Proc 34: 1468, 1975; Stroke 1981;12:653; K. Nakai et al. Scannig Microscopy 1989; Anat Rec. 1998;251:87; American Journal of Anatomy.1978;153:523; J Neurosurg. 2013;118:763; a discussion of potential role of sphincters in CSD can be found in: Neuroimage. 2011;56:1001). The functional impact of these structures on capillary perfusion were articulated in the arteriolar “stopcock theory” (Neuroimage. 2011;56:1001). Therefore, the conceptual novelty of the paper is limited.

Thank you for the comment, we agree and also point out that the “vascular sphincters” and “precapillary sphincters” have been reported previously (see below). However, the existence of precapillary sphincters in the cerebral vasculature has remained highly controversial (as documented by a recent review by Sakai & Hosoyamada (Ref. 14), claiming that sphincters do not exist in the brain) and their *functional role has not previously been described*. Hence, the conceptual novelty of the paper is 1) the confirmation of presence of cortical precapillary sphincters, 2) the characterization of their location, and most importantly 3) the description of their functional role.

The structure has not been described in most if not all reviews of the neurovascular unit. In line 40-41 we write: “Though precapillary sphincters have been known for decades, their existence except within the mesentery remains controversial”, and in line 239-241 we write: “Precapillary sphincters represent important anatomical sites of blood flow regulation due to their strategic placement at branch points of proximal PAs, where they regulate perfusion along the PA”. In comparison, “vascular sphincters” have been described as structures along the capillary walls and at capillary branch points (Cauli & Hamel 2004). Our work stresses the importance of distinguishing between the general term vascular sphincters and precapillary sphincters as the latter is an indentation of the capillary encircled by a mural cell *where it emerges from the penetrating arteriole*. Hence, the precapillary sphincter is strategically placed to adjust blood flow in both downstream capillaries and along the PA.

Regarding CSD and the “stopcock theory”, the paper referenced (Neuroimage. 2011;56:1001) explicitly states that their data do *not* support the notion of capillary sphincters *nor* the stopcock theory: “These

observations are in conflict with the historical arteriolar stopcock theory (Mchedlishvili,1986)”. The papers published by Tomita et al. and Mchedlishvili are excellent, but due to technical limitations they were not able to fully resolve the microvascular events in their studies as we have been able to do taking advantage of recent methodological developments.

We would also like to highlight that we have found several examples of precapillary sphincters (marked by blue arrows) and bulbs in prior cerebrovascular research articles (including our own, Hall et al. 2014) that neither mention their presence nor their role in blood flow distribution:

Peppiatt et al. (Nature. 2006 Oct 12;443(7112):700-4):

[REDACTED]

Hall et al 2014 (Nature. 2014 Apr 3;508(7494):55-60):

[REDACTED]

Longden et al. (Nature Neuroscience volume 20, pages 717–726 (2017)):

[REDACTED]

Finally, a comparison of the literature that mention precapillary sphincters to the literature that mention pericytes indicates that precapillary sphincters are an under-studied feature of the microcirculation:

2. The morphological, cellular and molecular description of these structures is also limited, and several questions remain to be answered:

(a) How frequent are these sphincters? Are they located on terminal segment of all penetrating arterioles and only a fraction of them? This information is critical because it will give a sense of their potential to impact the flow at the regional level and to protect the capillary bed, as suggested in the paper.

We agree that the frequencies and locations of the precapillary sphincters and bulbs are important and therefore the morphological and topological information is provided in Figure 2b-f (measured from as many penetrating arterioles as possible in 8 mice). In order to quantify the frequency, we had to make a criterion for the positive presence of sphincter or bulb at a branch point: sphincter <0.8 and bulb >1.25 times the diameter of a 1st order capillary (as stated in methods section). With these criteria, we found that precapillary sphincters and bulbs locate mostly to the upper layers of the cortex and mostly at the first couple of (i.e. proximal and *not* terminal) branches of the penetrating arteriole. Furthermore, Extended data figure 3 also gives an indication of the precapillary sphincter and bulb abundance in awake mice. To better accommodate the reviewer, we have revised the layout of Figure 2 to improve readability.

(b) Are there actually cells with contractile proteins located at the stricture? If so, what is their molecular signature? NG2 is present near the stricture, but NG2, a proteoglycan, is a non-specific marker which happens to be also in smooth muscle cells and pericytes, with little relevance to the purported function of these structures.

Yes, the cell encircling the indentation at the PA branchpoint, i.e. the precapillary sphincter cell, express smooth muscle actin (shown in Fig 3a, Extended data figure 4a and Supplementary video 2). With high magnification (Fig 3a and Supplementary video 2) we can even see the fibrillary structures of the actin overlapping with the NG2-dsRed signal. For these experiments, we applied the Betsholtz lab protocol (personal communication) with the FITC conjugated smooth muscle actin antibody in an NG2-dsRed mouse intravascularly stained by tomato-lectin and co-stained by DAPI. NG2 is a commonly used marker of oligodendrocytes and mural cells, and the purpose of using this transgenic mouse was solely to demarcate all mural cells of the vasculature. Please keep in mind that dsRed in the NG2-dsRed mouse is cytosolic.

(c) Are the cellular elements in question embedded in the endothelial basement membrane, like pericytes, or are outside of it, like smooth muscle cells? What are their ultrastructural characteristics?

In Figure 4d+e, we show that the sphincter encircling cell is embedded in between the endothelial basement membrane (tunica intima, where elastin is also present) and the tunica externa (where collagen alpha I type I is also present), where both capillary pericytes, smooth muscle hybrids on arterioles and smooth muscle cells of small arteries were also found. As we write (line 142-145): “Elastin was observed in the tunica intima of PAs and at the precapillary sphincter, but not in capillaries (Fig. 4d). Collagen $\alpha 1$ type-I staining was observed in the tunica externa of arterioles, precapillary sphincters, capillaries (Fig. 4e), and venules.”. We have not yet characterized the ultrastructure of the cerebral precapillary sphincter using electron microscopy because the main purpose and novelty of the present manuscript is to characterize and show the functional importance of the sphincter. Because the existence of a brain sphincter has previously been reported, we find that ultrastructural characterization is outside the scope of the present manuscript. To further investigate the localization of the precapillary sphincter regarding the basement membrane, we stained for the basement membrane collagen, collagen alpha I type IV (Supplementary Figure 7):

This shows the precapillary sphincter embedded in the basement membrane, like all other mural cells on the PA and capillaries.

(d) Elastin, COL1A1 and lectins are abundant in the cerebral vasculature. Smooth muscle actin may help with the identification of the cells and suggest contractile function, but the evidence presented is not convincing, the immunolabel seemingly being located also in endothelial cells (fig. 3a).

Thank you for the concern. As stated above, in Figure 3a (as well as Extended Data Figure 4a and Supplementary video 2) we have used a FITC-conjugated α -SMA antibody to mark smooth muscle actin. The α -SMA label overlaps completely with the NG2-dsRed staining in all our immuno-work. For Fig. 3a, there is a strong α -SMA and dsRed signal inside the sphincter cell encircling the lumen. Therefore, the signal does *not* arise from the endothelium – the seemingly low signal in the middle of the sphincter cell is due to the presence of the large nucleus. We have marked the borders of the precapillary sphincter cell (red) and the lumen (cyan) by dashed lines in the three grayscale images in Fig. 3a to document this fact (lines 684 - 685), where the endothelial cell is situated between the inner red dashed line and the cyan dashed line and has absolutely no α -SMA staining:

(e) There have been major advances in the molecular signature of mural cells, and a number of useful markers have been identified (Nature 2018;80:475; Nat Neurosci. 2017;79:559; Sci Rep. 2016;6:35108). This new knowledge could be used to provide a better understanding of the cellular identity and nature of these structures.

We thank the reviewer for the suggested markers of mural cells and pericytes. We have contacted the Betsholtz lab to inquire the most relevant antibodies to differentiate between pericytes and smooth muscle cells, their reply was the following:

“Unfortunately, neither of [Vitronectin and PDGFRB] are more specific to pericytes. If you look at our published database (<http://betsholtzlab.org/VascularSingleCells/database.html>), then you see that all the pericyte markers Anpep (Cd13), Pdgfrb, Vtn and Des are also expressed in venous and arterial smooth muscle cells. So whichever pericyte marker you choose, it would be good to stain also for Acta2, to exclude the arterial smooth muscle cells. Unfortunately, venous smooth muscle cells (bigger diameter vessels w/o Asma staining) and pericytes are very similar to marker expression, although the morphology is slightly different. It would be helpful to include the CD31 staining to be able to see the vessel diameter and in that way decide if the mural cell is pericyte (on the capillaries) or smooth muscle cell (on the venules). I prefer to use the Anpep antibody, because it works the best out of all the pericyte antibodies.”

This means that the Betsholtz lab defines pericytes as non-contractile mural cells situated on capillaries (and not venules), which is the definition also used by Grutzendler lab (Hill et al. Neuron. 2015 July 1; 87(1): 95–110.) but different from the definition used by many other labs (Hall et al. Nature. 2014 Apr 3;508(7494):55–60., Grant et al. J Cereb Blood Flow Metab. 2019 Mar;39(3):411–425., Rungta et al. Neuron. 2018 Jul 25;99(2):362–375.e4.), who include contractile mural cells on the 1st-4th order capillaries as pericytes (ensheathing pericytes and mesh-like pericytes). Betsholtz lab favorite pericyte marker Anpep (CD13) is not specific for pericytes (PC), but is also present in venous smooth muscle cells (vSMC), arteriolar smooth muscle cells (aaSMC) and two types of vascular fibroblasts-like cells (FB):

We have now tested two pericyte markers CD13 and CD146 (Supplementary Figure 7) and found that neither of them is specific to pericytes. We confirm that CD13 exists on string/mesh-like pericytes, but also on the ensheathing pericytes, arteriolar mural cells, venous smooth muscle cells and pial fibroblasts. We also confirm CD13 expression on an NG2-dsRed negative cell type on the PA, likely the fibroblasts-like cells that Betsholtz has described. The precapillary sphincter has a coverage of CD13 like the neighboring ensheathing pericytes and arteriolar mural cells.

CD146 was expressed mainly in endothelial cells, and we found a weak staining in arteriolar mural cells (white arrow), but no staining in pericytes or the precapillary sphincter:

Unfortunately, the batch number of vitronectin used by the Betsholtz lab is no longer available, and when we tested a newer batch, it did not show specific staining.

With regard to the Neurotrace 500/525 marker, we have tested it *in vivo* and find that it is not specific for mural cells but also marks a population of cells in the pia (possibly macrophages), here shown as a maximal projection of the first 50 μm of the pia of an NG2-dsRed mouse with Neurotrace 500/525 (1:25 dilution) applied for 5 minutes, washed out and imaged 1-4h later (note that there is some bleed-through from the green to red channel, but importantly not the other way around):

Here we show the emission spectra of the Cascade Blue, Neurotrace 500/525 and dsRed, as well as the bandpass filters for the three channels used, which allowed us to separate blue from green from red. To visualize vessel lumen, we either i.v. injected Cascade Blue dye or FITC (but the latter subsequent to Nissl Neurotrace 500/525 imaging):

[REDACTED]

We find neurotrace 500/525 signal in string-pericytes (Supplementary Figure 7), like what was reported by Damisah et al.:

However, also ensheathing pericytes and arteriolar SMC/pericyte hybrids contain signal of the marker (although with less intensity), here shown before (left) and after FITC i.v. injection (right) – notice how the mural cells on sphincter and PA are orange on the left image (green+red overlay). Contrary, the right image shows the FITC signal overpowering the Neurotrace 500/525 signal and they appear red:

Thus, we do not find this sharp demarcation between “precapillary mural cells” and string-pericytes as reported by Damisah et al. and do not consider Nissl Neurotrace 500/525 a specific marker for non-contractile pericytes.

To conclude, we do not believe that the evidence supports that the precapillary sphincter is a new cell type different from vascular smooth muscle cells or pericytes. Rather, it is a contractile mural cell within the spectrum ranging from spindle-shaped arterial smooth muscle up to the non-contractile pericyte on higher order capillaries. We would again like to stress that our main purpose is to show the strategic importance of the precapillary sphincter for flow distribution and protection of downstream capillary beds.

3. The data show dilatation of these structures during neural activity and exposure to papaverine. However, these manipulations will also dilate the arteriole in continuity with these structures. Therefore, the observed dilatation and constriction could simply represent the diameter change of the terminal segment of the parent arteriole, and the size modification of the stricture could be related to mechanical forces transmitted from the arteriolar system upstream. The same could be said of the changes in size observed with CSD or cardiac arrest.

This is an important point, which we have thought about extensively, but it is extremely hard to target experimentally. As the sphincter encircling cell is intimately connected to both the 1st order capillary and particularly the relatively large PA, we obviously cannot rule out a contribution of PA vasoactivity to the vasoactivity of the sphincter. However, we show that the mural cells on the 1st order capillary, including the sphincter, contain α -SMA and that both the capillary and the PA have a tonus in the resting state. While passive force transmission from the neighboring segments is very likely to influence the sphincter, it seems unlikely that the α -SMA containing sphincter cell is entirely passive. As in every other contractile vessel segment, it is reasonable that the contractile cell circumscribing the relevant segment exerts a relatively high level of influence on the tone of the segment. Yet, no matter what vessel segments exert the highest level of influence on the tonus at the precapillary sphincter, the indentation remains a bottleneck for blood flow to the capillary bed at resting state, during dilation and during constriction, i.e. the functional relevance of the sphincter remains intact. We have now added a couple of sentences in the discussion to accommodate the comments of the reviewer (line 225 to 230).

4. The impact of the observations of single capillaries, presented in the paper and in the movies, on the overall tissue perfusion and purported protection of capillaries remains to be established and will depend on how frequent these structures are.

As stated above, we would like to draw your attention to Figure 2b-f, where we have quantified the presence of precapillary sphincters and bulbs in the different cortical layers and at increasing PA branch point

numbers. In addition, we have now included a mathematical model that provides a quantitative estimate of the level of adverse pressure protection the sphincter provides to the downstream capillaries of proximal branches of the PA. The mathematical model is based on a reconstructed PA network and clearly demonstrates that the sphincter bottleneck is required to lower the blood pressure in the proximal capillary branches of the PA, particularly if the PA is large and hence, is expected to carry a large pressure at the inlet (see Fig.2 and Fig.6 as well as the section on model simulations from line 170).

5. Although the data on CSD and cardiac arrest are of potential interest, the findings are descriptive and anticipated based on previous literature, and there are no experiments on the mechanisms regulating the size of the sphincters and on their impact on the pathological changes that could derive from their engagement.

We appreciate the reviewer's consideration of relevance and have moved the experiments on cardiac arrest to the supplement. We have kept the CSD experiment (Figure 5) in the manuscript, as we believe that it is an important observation that during phase III of a CSD, the flow resistance at the sphincter increases meaningfully and up to a degree where blood cells can temporarily stall (Supplementary video 6). An important point here is that the effect on flow resistance of the general constriction observed in phase III of a CSD is very high at the sphincter compared to other segments. Thus, the downstream capillaries receive a compromised perfusion. This points to a structural deficit of the topological layout of the cerebrovasculature: While the sphincters are important for maintaining acceptable levels of pressure (and redistribute perfusion) under normal conditions, they may become problematic during adverse pathological conditions that promotes a generally constrictive environment.

Reviewer #2 (Remarks to the Author):

Rebuttals to reviewer 2 are written in red

In this manuscript, Grubb et al. provide novel findings in the cortical microvasculature of the brain that identify "precapillary sphincters" between penetrating arterioles and the 1st order vessel. Using live imaging with 2-photon microscopy and immunostaining on brain slices, the authors provide an elegant and convincing structural and functional characterization of this vascular segment at baseline, during functional hyperemia and pathological conditions. The authors conclude that these sphincters represent a mechanism to equalize perfusion to capillary beds along the length of penetrating arteries (PA) and to protect brain tissue against adverse blood pressure change.

This is a novel and interesting study that may be of interest to others in the community. Overall the data shown are solid. However, there are weaknesses and other issues that need to be addressed.

1) Results shown on figure 2 are unclear and confusing.

Thank you for pointing out the areas of confusion. We have clarified the figure according to the specific issues.

a) The terminology "PA number" and "1st order capillary" is confusing. In particular, controversy has surrounded the definition of capillary versus arteriole, and pericytes versus smooth muscle cells.

We have now included the PA branch point numbering in Figure 2a that we use in the rest of Figure 2. Importantly, our convention for the PA numbering is not part of the ongoing controversy regarding

terminology. First, the term “PA branch number” is replaced with “PA branch point number” and the relevant y-scale in Figure 1c shows “#1” to “#7”. Second, to stay clear of confusion regarding the definition of capillaries vs arterioles, our lab has adopted an agnostic approach based on branching order. Specifically, a PA branches into 1st order capillaries. 1st order capillaries branch into 2nd order capillaries. 2nd order capillaries branch into 3rd order capillaries etc. A new version of Figure 1e show this convention and additionally illustrates the smooth muscle to pericyte continuum as an aid to resolve the controversy. We have kept the term “precapillary sphincter” due to the historical use of this particular nomenclature.

b) Despite the numerous metrics displayed in this figure, some issues are still unclear. Do these sphincters occur only at the boundary between the PA and its 1st branch. As suggested by the authors, the mouse cortex is supplied by short and long cortical arteries. Does the distribution of sphincters differs between short and long cortical arteries.

Yes, the precapillary sphincters are defined as the “...as dsRed-positive cells encircling an indentation of the lumen at PA branch points” (see also the new Figure 1e). As noted in the discussion, hotspots of local constriction within arterioles and capillaries have been described before (see Cauli et al 2004, ref 41) but the precapillary sphincters are unique due to their particularly strategic location. We have now revised and clarified the definition (line 47).

As noted in the next comment, our imaging depth was confined to a cortical range from 0 to ~700µm. For this reason, we unfortunately could not resolve the depth of most PAs. From our data in Fig.2, we observed most sphincters on the larger PAs that penetrated deep into the cortex, so we anticipate that longer PAs also correlate with a higher incidence of sphincters as longer PAs – all else equal – need a higher driving force to propel blood into the deep cortical capillaries.

c) The imaging depth of 2-PM is usually limited to 0-600 µm in the cortex. Was the analysis of the localization of sphincters within the cortex performed *ex vivo* or *in vivo*?

Actually, we analyzed both *ex vivo* (coronal brain slices) and *in vivo* data with similar results, but the data set from the analysis presented in Fig. 2 was derived exclusively from *in vivo* imaging. In the mice dedicated to this subset of experiments, we imaged as many penetrating arterioles as possible and as deep into the cortex as possible. The disadvantage of *ex vivo* analysis is that there is no blood pressure, and therefore the vessel lumen (including the precapillary sphincter) is underestimated – the advantage is that there is no depth limitation. In both cases, we found highest incidence of precapillary sphincters and bulbs in at the first couple of branch points in the upper layer of the cortex.

2) On figure 3, it is unclear how flow resistance in the PA, sphincter, bulb and 1st capillary was estimated. What do the authors mean by “proxy resistance”. Based on the measured RBC velocity and the Poiseuille’s law, the authors elaborate on the pressure drop in the sphincter and the 1st order capillary. This needs to be further explained, especially during whisker stimulation where both velocity and diameter increase in the sphincter. In general, panel 2g and this paragraph are hard to follow.

We thank for reviewer for pointing out the lack of clear description. By “proxy resistance” we simply mean the flow resistance calculated from Poiseuille’s law ($\Delta P = \frac{8\mu L Q}{\pi r^4}$, where P is pressure, μ is dynamic viscosity, L is length, Q is flow, and r is vessel radius) using the standard hemodynamics measure of flow resistance (R) in laminar fluid flow (Ohm’s law $\Delta P = RQ$), i.e. $R = \frac{8\mu L}{\pi r^4}$. We do not know the specific dynamic viscosity but it is reasonable to assume that it is the same between the PA and the group of Sphincter, Bulb and 1st order capillary. As 8 and π are just constants, our “resistance proxy” per unit length ($R_{proxy} = R \frac{\pi}{8\mu L} = \frac{k \cdot R}{L}$, where k is a constant) therefore becomes $R_{proxy} = \frac{1}{r^4}$. The same reasoning underlies panel 2g,

where we revised and clarify the calculation in the legend (see line 673-674). The description of the “resistance proxy” has now been included in the “Methods” section (see lines 393-398).

3) The authors analyzed vasomotor responses of the sphincters during prolonged cardiac arrest (> 20 minutes), which relevance is highly questionable. Hence, I would recommend to delete this experiment and rephrase the title accordingly. Instead, given the proposed role of these sphincters in the protection from high pressures and the perfect in vivo imaging model to analyze this, the vasomotor response of these sphincters to acute blood pressure increase or decrease, ie in the context of cerebral flow autoregulation, must be assessed.

We appreciate the reviewer’s consideration of relevance and have moved the experiments on cardiac arrest to the supplement. Regarding the role of sphincters “in the protection from high pressure”, first we note that sphincters protect *downstream* capillaries against adverse pressure (see Fig. 6). Because the PA represents a “one-way-street” for blood to enter capillary beds within the cortex, the blood pressure and RBC flux is higher at the proximal end compared to the distal end. If capillaries branching off the PA were identical, the proximal capillaries would experience a higher pressure and higher RBC flux compared to the lower capillaries. This consideration would not differ with changing systemic pressures. Second, changes in systemic pressure stimulates regulation of heart rate, stroke volume, and sympathetic output (which in turn causes secondary impact on flow-mediated responses) and thus, systemic tone regulation. These overall changes in baseline makes it difficult to interpret the effect of a locally imaged sphincter in the context of acute cerebral flow autoregulation. We have tried to accommodate the reviewer via a set of preliminary experiments, where blood pressure was increased by a bolus injection of α -chloralose. Here, we do find that the PA, sphincter and 1st order capillary constricts (after a short initial dilation):

Yet, the proper controls and study design warranted by this question requires a complete study and is well beyond the scope of this first paper on the functional role of precapillary sphincters. However, we do intend to follow up on this idea.

To avoid inadvertent extrapolations of our conclusion regarding the sphincter's capacity of "pressure-protection", we have refined our wording where appropriate throughout the MS (e.g. line 19).

4) A mathematical modeling of pressure changes from the PA to the first order vessel at baseline and upon blood pressure changes would strengthen the conclusion that the sphincter complex protects capillaries from blood pressure peaks.

We acknowledge the idea of strengthening our conclusion by developing a mathematical model describing flow and pressure changes in a small cortical network and have done so accordingly. The results are shown in Fig. 6 and described in the "Results" section (lines 174 to 198). A section in "Methods" (lines 400 to 435) describes the model construction and assumptions.

5) The repeated statements throughout the Results section that precapillary sphincters "contribute to pressure distribution", "play an active role in the protection of downstream capillary" etc.. should be deleted and reserved to the discussion.

We acknowledge the improper use of wording in the "Results" section and have revised the text accordingly (see lines 67, 113, 126, 147, and 153). Thanks.

Reviewer #3 (Remarks to the Author):

Rebuttals to reviewer 3 are written in blue.

Grubb and colleagues have investigated the role of a specific cell on the vascular tree called the precapillary sphincter and how it could regulate blood flow. They nicely use the NG2-DsRed mouse, vascular tracing and two-photon imaging to visualize the sphincter and provide a clear definition as to what constitutes the sphincter. They show that during whisker stimulation in mice that the sphincter can relax leading to an increase in blood flow to the downstream capillary bed and that surrounding the sphincter were structural elements such as elastin and collagen to protect the brain against pressure-induced injury. The papaverine experiments were technically demanding but well performed. Subsequently, the authors go on to show that the sphincter contracts during both induced spreading depolarisations and following cardiac arrest which would limit the capillary blood flow in these conditions. Overall, this study utilizes an excellent imaging setup to visualize changes in the pre-capillary sphincter during neurovascular coupling and pathology, ultimately influencing blood flow.

The techniques used in this manuscript are sound, highly validated and are suitable for the aims of the manuscript. The conclusions are supported by the evidence, and the use of in vivo techniques were appropriate and strengthened these conclusions. With regards to statistics, most of the statistical analysis appears correct, but no apparent power calculation was performed to determine sample sizes.

The role and even the presence of a precapillary sphincter in the cerebrovasculature is still controversial. This novel manuscript sheds some light on the existence of these structures and their influence on blood flow and also adds to the debate about the role of pericytes vs vascular smooth muscle cells on controlling cerebral blood flow.

Specific comments below:

1. Given the placement of the precapillary sphincter at a branch point of a penetrating arteriole and the first order capillary, they are placed in an ideal position to influence capillary blood flow. However, is the name “precapillary sphincter”, a misnomer? Are these in fact just contractile pericytes or ensheathing pericytes as they are now known? What is the difference between the action of the so-called precapillary sphincter to the action of the downstream capillary pericytes which actively dilate the blood vessel upon neuronal stimulation? Some discussion was provided on lines 239-243 about the naming of this. Some greater elaboration and discussion about the NG2+ α -SMA+ cells at the sphincter is required and whether they are just a VSMC or a pericyte or a hybrid cell, and how this specific cell type fits in with the continuum of VSMCs to pericytes.

We appreciate the reviewers request for additional discussion. The primary reason for naming this structure differently from other vascular structures is – as the reviewer acknowledges – the strategic positioning at the initial part of the 1st order capillary that emerges from the PA. The ability to actively regulate the “bottleneck structure” at the branch point has functional impact on blood flow as increased flow resistance across the sphincter reduces blood flow and increases plasma skimming (see Fig. 6) which combined tend to equalize perfusion along the PA. While changes in flow resistance can also be achieved by downstream contractile capillary pericytes, the benefits of placing the bottleneck at the branch point are: 1) reduced pressure across the *entire* capillary, thereby 2) enabling the existence of the Bulb, which has an inherently higher risk of rupture due to the larger diameter (and apparently lower structural embedding). Finally 3) plasma skimming is not effective by placement of a sphincter at a downstream capillary site. We have included short paragraphs in the “Discussion” section (lines 206-209 and 257-260) that elaborate on these differences.

While we present evidence that the sphincter cell contains α -SMA, is actively contractile, and of mural origin, we focus on the *function* of the cell and believe that labeling of the cell (SMC, pericyte or hybrid) is of secondary importance. However, we do *believe* that the sphincter cell is a hybrid due to its morphological characteristics but do not have conclusive evidence as to provide a clear-cut label. However, to accommodate the reviewer, we have provided additional cellular and molecular characterization of the sphincter cell (lines 59-61 and Supplementary Figure 7). However, specific markers for the pericyte remains to be found.

2. Line 57, “close inspection revealed that PAs were covered with VSMC/pericyte hybrids (Fig 1e)”. I agree with the authors in that there is a continuum of mural cell cyto-architecture from pial arterioles to 3rd order capillaries. But I disagree that the mural cells covering PAs are VSMC/pericyte hybrids, they should be referred to as VSMCs as that is what the imaging suggests and the depiction in Fig 1E is misleading with a slight protrusion of cell bodies on the PAs.

We thank the reviewer for the comment and have revised the manuscript accordingly (line 57-58).

Regarding mural cell cyto-architecture, we refer to the beautiful work by K. W. Zimmermann (1923) who made detailed drawings of mural cells on the transition from arteriole to capillaries in cortex, which are the most precise illustrations we have seen:

Arteriole (20-16.7 \$\mu\$ m)

[REDACTED]

Precapillary (14.5 μm) to capillaries:

[REDACTED]

Also, the illustrations by August Krogh (1929) indicates rounded cell bodies of the mural cells on the end-arteriole, and ensheathing pericytes on large capillaries:

[REDACTED]

Scanning electron microscopy (by Don W. Fawcett) has revealed that string like pericytes (A) are not merely strings on top of the capillary, but they reach around it with fine processes. Ensheathing pericytes (B) have many arms that reach around the capillary over a shorter distance and the (small) arteriole (C) has mural cells with rounded cell bodies and many arms reaching around the vessel:

[REDACTED]

To improve our illustration of the *morphological* features of mural cells on the PA and capillaries, we based it on confocal microscopy data of DAPI stained coronal slices (left) of an NG2-dsRed mouse and carefully drew the outlines of each cell as well as their nucleus while changing the focal plane to follow the structures (Figure 1e, right). Here, the PA is 15-18 μm in lumen diameter at the precapillary sphincter, which is slightly above the average PA lumen diameter we have observed *in vivo* ($11.4 \pm 0.6 \mu\text{m}$) but it is a good representation of the mural cell morphologies we observe:

For comparison, we show below the morphological features of large pial arteriole smooth muscle cells (red: NG2-dsRed, blue: DAPI – notice the classical spindle-shape and the elongated nucleus of the SMC):

The morphological differences between SMCs located on large arterioles, on the penetrating arterioles, and veins are clear. Therefore, to respect the works by Zimmermann, Krogh and Fawcett among others, we have strived to make as precise an illustration as possible that describes the spectrum of cellular morphologies.

3. The definition of the sphincter is crucial for the analysis. It is clearly stated that a sphincter is defined by DsRed-positive cells encircling an indentation of the lumen at PA branch points and usually followed by distention of the lumen (the bulb). However, if there is no bulb is it considered a sphincter? The analyses presented in Fig 2 suggest that <50% of branch points off the PA had a sphincter or a bulb. Then how important is the sphincter to overall changes in flow if the sphincter/bulb is not present at the majority of branch points. At the branch points with no bulb, were there DsRed-positive cells where the sphincter would be? Do these vessels dilate in response to stimulation? Some data here on these vessels without sphincters would be informative and some discussion around why only the minority of branch points would have sphincters is needed.

We thank the reviewer for raising these clarifying questions. First, the critical words in our definition was "...usually followed by a distention..." As such, the sphincter is the indentation (with or without the Bulb) and the indentation is circumscribed by a mural "sphincter" cell. Second, while it is true that most branch points did not have a sphincter, it is also true that almost all PAs had one or more sphincters and that those are located primarily at the proximal bifurcations of larger PAs to larger branches (Fig.2). Our explanation is therefore, that these larger and proximal "high pressure" PAs need just one or more sphincters to equalize the perfusion to the capillary beds that branch off the PA as it dives deeper into the cortex (if all had sphincters the effect of equalizing perfusion along the PA would be entirely lost). The number of sphincters required (and their individual level of contraction) in turn depends on the size and length of the PA (and 1st order capillaries) as well as on the number of branches from the PA. Note also, that our analysis therefore is sensitive to our definition of the sphincter. A less strict requirement (say, 10% constriction) would obviously entail more branches labeled as sphincters and vice versa. Third, as our aim of the paper was to describe the precapillary sphincter in health and disease, we did not embark on a quantitative analysis of branch points *without* sphincters. In general, the functional characterization of 1st order capillaries (mostly without sphincters) are already described by the literature, see e.g. Cai et al. PNAS 115(25) 2018.

We have revised our definition (line 46-49): "...as DsRed-positive cells encircling an indentation of the lumen at PA branch points (Fig. 1a). Precapillary sphincters were often but not always followed by a distention of the lumen, which we denoted as "the bulb"." Also, we have extended our discussion around why only a minority of branch points have sphincters (line 192-194).

4. Hall et al 2014 Nature showed in their landmark paper that capillaries dilated before PAs in response to whisker stimulation. In terms of onset of dilation post-stimulation, how did the sphincter dilation compare to the timing of PA and 1st order capillary dilation? If it is delayed, could this then just be a passive response to the dilation from the capillary and the PA?

We did carry out experiments that investigated the latency between dilation of the different segments. Unfortunately, the PA, sphincter, and 1st order capillary segments are too close and the recording speed too slow to permit a conclusive (rigorous, and reproducible) separation of the timings. We do not believe that the vasomotor responses are entirely passive as this would leave out any reason for having a large α -SMA expressing sphincter cell around the highly vasoactive sphincter region (Fig.3 and in Fig.4 we show that the sphincter has a larger capacity for dilation in absolute diameter compared to the 1st order capillary). Yet, we cannot rule out that part of the sphincter vasoreactivity is mechanically transferred by a vasomotor response in the PA. Functionally, it is mostly of secondary importance to what proportion the sphincter is passively influenced by the PA (in cases of antagonistic responses, the 1st order capillary and the sphincter most likely reinforce each other to overcome the mechanical influence of the larger PA).

5. Line 161 states that in phase II the sphincter had greater dilation (39±8%) than the bulb and 1st order capillary (22±3% and 21±4% respectively). However, these data do not match the dilation represented in the graph in Fig 5d (sphincter looks around 31%, bulb looks around 20% and 1st order cap. looks around 18%). Please clarify this inconsistency.

We thank the reviewer and fully understand reason for confusion here. As we state in the legend of Fig.5, the phase II data was log-transformed to ensure homoscedasticity in the LME analysis. Yet, in the manuscript, we stated the mean±/SEM for the untransformed data in order to be directly comparable with the maximal dilation during whisker stimulation and papaverine application. Hence, the different numeric values. To avoid confusion, we revised the MS to just show the statistical comparison of the 3 independent groups (using a Kruskal-Wallis test) while removing the characterizations of the distributions (see line 165).

6. The representative figure in Fig 5c shows oligemia out to 300 seconds after the onset of depolarisations with KAc. How long-lasting is the sphincter constriction during phase III beyond 300 seconds and when

does this resolve itself? Are the resistance calculations in Fig 5e determined from the minimum diameter change during phase III, or incorporate a measurement of diameter change over time to calculate a chronic resistance change, as this is not clear.

The oligemia in phase III extends at least 30 min and up to 45-60min (Khennouf et al, Brain 141, 2018; Ayata & Lauritzen, Physiol Rev 95, 2015). We did not particularly image the sphincter region for this long duration and can only confirm that the sphincter remains contracted for up to 10 min, i.e. spanning the duration of the present experiments. However, given the entire 1st order capillary remains constricted for more than 30 min (Khennouf et al, Brain 141, 2018) and that the sphincter is located on the initial part of the 1st order capillary, we believe it is highly likely that the sphincter along with the rest of the capillary remains constricted during phase III.

Regarding the resistance calc in Fig. 5e, we thank the reviewer for pointing out this confusion. Actually, we had calculated the resistance based on the minimum diameter but this is a minor mistake, given that the minimum value should be treated as a “random value” (random in the sense that it is a sample and not a defining characteristic of a distribution). Therefore, we have changed the calculation to the mean diameter value during our recording of phase III. See new sub-figure 5e (line 735).

7. The methods for the cardiac arrest experiments are not well described. The methods state that mice were euthanized by iv injection of pentobarbital but nothing else was mentioned about imaging beyond then. Fig 6 shows that vascular changes were gradual with a final collapse around 25mins following pentobarbital administration. Do you have heart rate data to show that at what stage of the pento delivery was heart rate stopped, was it immediately? Also, the sphincter data would be related to the massive blood pressure drop from cardiac arrest which affects the PA which goes on to effect the sphincter and downstream capillaries. There is no description of an active sphincter constriction, just a delayed one due to presumably the blood pressure drop. Therefore, I am not really sure what these data are showing about specific sphincter biology relative to the PA and capillaries and I am not sure adds much to the manuscript. However, what would be interesting if this is a reversible effect (think of cardiac arrest patients who are resuscitated) and how the sphincter would react then with a restoration of blood flow? Do they remain constricted as described in the pericyte literature during reperfusion following stroke which would limit blood flow to the capillary bed? This would be of much greater interest to the field.

We do have the heart rate data and know that the heart stopped within seconds from pentobarbital administration. However, we acknowledge the fundamental weakness in observing changes in an irreversibly a dying animal and have opted to move the reporting of this experiment to the supplement. The primary take away from the experiment was the eventual collapse of the sphincter demonstrating the vulnerability of a bottleneck. Like the reviewer, we believe that a better model would be a reversible cardiac arrest (or stroke). The experiment was replaced by a mathematical model of the sphincter’s role in pressure protection of downstream capillaries.

8. Line 200-1 states that “we observed α -SMA along the PA and 1st-4th order capillaries. However, Ext data fig 4b second panel doesn’t show any α -SMA labelling in 4th order capillary (in fact this capillary based on the scale is around 10 μ m which is very large for a 4th order capillary). The large panel in Ext data fig 4a shows α -SMA labelling only down to the 2nd order. This needs to be corrected in the text and the title of extended data figure 4.

Thank you for the concern. We agree that data in Sup. Fig.4a does not show α -SMA labelling in 4th order capillary. Yet, in many cases, α -SMA was observed on 3rd order capillaries. To clarify, we have changed the sentence to: “we observed α -SMA along the PA and in some cases up until 4th order capillaries”.

9. Both male and female mice were used as well as a large age range, could these have influenced outcome?

We strongly believe that the use of both sexes and a fairly large age range *underscores* our conclusion that precapillary sphincters are present and have functional impact (we have also found them in aging animals). At the onset of detailed investigation, we thought that employing only one sex at a particular age would diminish the robustness of the findings. Also, given the similarity of our findings in both sexes and across ages, the use of a single sex at a particular age would probably not have influenced the outcomes to any significant degree.

10. Line 277, “to ensure that the precapillary sphincters were not a result of the craniotomy”. I am not sure how a craniotomy would cause a precapillary sphincter, but it is important to perform some experiments with an intact skull to overcome any effects of a craniotomy.

We fully agree with the reviewer but opted to include the control experiment due to several questions raised from colleagues.

11. Line 326, “mice that did not exhibit normal vessel diameter responses to stimulation were discarded”. Please define what is normal, is it a certain % increase in diameter? Please add the criterion to the methods.

We have now included the criterion in the Method section (line 348): “...mice that did not exhibit normal vessel diameter dilation, i.e. less than 5% dilation, to whisker stimulation were discarded...”

12. Is reference 30 the same as reference 18?

Yes, we are grateful for catch!

13. In Fig 3, * are used to represent statistical significance but this is not mentioned in the panel or the legend.

Thanks for the catch. The relevant information is now included in the legend (line 711).

14. Extended data table 1, need to specify the n number of mice and vessels analysed in this table.

Thank you for pointing this out, we have added the number of mice and vessels analyzed (line 778).

	N (animals)	n (vessels)
PA	13	19
Sphincter	13	19
Bulb	9	11
1 st order cap	12	18

15. Extended data figure 3 shows the awake mouse and the presence of the bulb adjacent to a sphincter. However, this is not a DsRed mouse and so the DsRed cells can not be visualized. It would be nice to have an awake NG2-DsRed mouse imaged to confirm that at each bulb, there is a DsRed+ cell at the adjacent sphincter.

We acknowledge that a DsRed mouse would have been nice to have. However, the awake mouse experiments were performed during a research stay at another lab. Thus, we could not use a DsRed mouse as the relevant lab did not have access to this mouse line. Unfortunately, we therefore cannot provide the image.

Reviewers' comments:

Reviewer #1 (Remarks to the Author):

The authors have addressed some of the concerns raised in the review. There are few issues remain to be dealt with:

Line 45-and following: The frequency of sphincters needs to be mentioned here. What would be the impact on tissue perfusion if the sphincters are present only in 50% of the arterioles?

Line 59-61: Shouldn't α SMA be included in this sentence?

Line 174: does the model assume that only 50% of the arterioles have sphincters? If these structures are protective, then some arterioles are unprotected and more susceptible to damage?

Line 263: If the sphincters are bottlenecks to capillary perfusion, how can the downstream capillaries regulate flow? Does the regulation of flow by capillaries occur only in the portion of the vascular network devoid of sphincters?

Figure 2: The designation of "positive" is confusing. On how many vessels and in how many mice was the assessment of number and location made? The "biophysics" was predicted according the Poiseuille's law and not assessed experimentally as the title of the figure implies.

Reviewer #2 (Remarks to the Author):

The manuscript has been improved. Authors have now included a mathematical model to support their conclusion. However, panel g (Fig. 6) must be clarified. How the model can estimate sphincter contraction?

Figure 2, which has been barely modified, is still unclear. How many PA have been analyzed, how many sphincters have been counted. These information are not in the data source. On panel g (left), characters are too small, what does the Y axis mean? Please clarify the legend.

Reviewer #3 (Remarks to the Author):

Grubb and colleagues have provided a thorough response letter to the reviewers' comments and have substantially modified the manuscript. I believe these revisions have significantly improved the manuscript. The authors have addressed all of my concerns. I only have a couple of minor comments to address:

1. Extended data Table 1 now has the n-number for vessels from N animals. However, the difference between n and N are not actually described in the table or its legend. Please include n = vessels and N = animals.
2. The new schematic in Fig 1E and Extended Data Fig 8 is excellent, and that it was drawn based on an actual image is even better. I think including the confocal image presented on p16 of the rebuttal letter should be included in Extended Data Fig 8 to show the accuracy of the schematic.

Reviewers' comments:

Reviewer #1 (Remarks to the Author):

The authors have addressed some of the concerns raised in the review. There are few issues remain to be dealt with:

Line 45-and following: The frequency of sphincters needs to be mentioned here. What would be the impact on tissue perfusion if the sphincters are present only in 50% of the arterioles?

Thank you for the suggestion and question, which has facilitated additional clarification of our paper. A “frequency of sphincters” can be interpreted in more ways:

Interpreted as the *proportion of PAs containing one or more sphincters*, the overall frequency is calculated to be 72%. This overall frequency masks a marked heterogeneity among PAs that arise from each serving different brain regions (see Fig. 2a); for example, some go deep with few proximal branches, some are short and have low caliber, while others branch extensively and serve the upper cortical layers. In addition, 1st order capillaries that almost immediately branch into two 2nd order capillaries have in-built “pressure-protection” as the pressure-head is reduced in the two branches, reducing the need for a sphincter. However, most PAs give rise to at least one, typically proximal, branch that contain a sphincter to ensure optimal perfusion to each capillary bed along the PA while protecting capillaries and soft brain tissue. We have included this consideration in the manuscript (line 65-67).

Interpreted as *the average proportion of sphincters found on branch points to the PAs*, the frequency is not a straightforward measure for the following reasons: 1) The specific number of sphincters and bulbs found depends on the criteria for identification, which is a somewhat arbitrary value that we defined as <0.8 times the diameter of a 1st order capillary for the sphincter (and for the bulb as >1.25 times the diameter of a 1st order capillary). Hence, the number of positively identified sphincters/bulbs will change with changing criteria. 2) The effect of highly diverse morphologies and topologies among PAs, are simply not considered if we just presented a single frequency of sphincters. Rather, we first defined a conservative criterion for sphincter/bulb inclusion that matched the distinction readily observed by eye from our images. Second, in Fig. 2 we highlighted a) the heterogeneity of PAs and therefore measured the occurrence of sphincters (or bulbs) with respect to b) cortical depth, c) number of branch points along the PA, d) the diameter of the PA, e) the diameter of the 1st order capillary. These graphs highlight the multiple factors that the occurrence and location of the sphincter depend on. As such, it would not be justifiable to present a single “frequency of sphincters” value. We have clarified these points (line 65-66 and Fig. 2).

With regard to the impact on tissue perfusion due to sphincter presence only in 50% of proximal arteriole branchpoints, we first note that tissue perfusion is a broad entity that depends on 1) the activities of the brain tissue perfused by the PA and 2) the aggregate morphological and functional properties of the cortical network of which, the sphincter represents a local part. Second, our data suggest that the presence of a sphincter, i.e. the indentation of the capillary where it emerges from the PA, is mainly resulting from a high pressure in the PA right at the branch point that would otherwise be damaging to capillaries; most likely at proximal branches. The heterogeneity among PAs, however, give rise to heterogeneity in sphincter location and occurrence as shown in Fig. 2. The heterogeneity and the multiple factors that determine tissue perfusion preclude a clear answer to the impact of tissue perfusion due to a presence in only 50% of proximal arteriole branchpoints. However, it should be noted that the strategic location of sphincters uniquely positions them as master regulators of the downstream capillary network perfusion.

Line 59-61: Shouldn't α SMA be included in this sentence?

We have now included a reference to the α -SMA staining here as well (line 61). We had deferred this part to the discussion of Fig. 3 but agree that this could cause confusion.

Line 174: does the model assume that only 50% of the arterioles have sphincters? If these structures are protective, then some arterioles are unprotected and more susceptible to damage?

Based on the image reconstruction of a specific PA, the model has incorporated sphincters in the two proximal branches of the PA (the locations have now been explicitly shown in Fig. 6a, right). For this PA, the proportion of sphincter containing branches is 2 out of 7 capillary branches in total. In Fig. 6b, we consider the effect of having the sphincters at the two PA branches (green curve) versus not having a sphincter (red curve). This PA only "needs" to have sphincters at the two topmost branches as the pressure has declined so much along the PA at the 3rd to 6th branch that sphincters are unnecessary for the health of these capillaries (see Fig.6a, right). In fact, if sphincters were to be found at those distal branches also, there would be no added benefit of pressure and RBC flux homogenization along the PA.

In short, sphincters are only necessary in branches where the pressure in the PA is high, i.e. at the proximal branches, and this uneven distribution of sphincters along the PA (proximal vs distal) helps to drive more perfusion toward the distal branches of the PA. In branches without sphincters, the pressure difference between the PA and the capillaries is not large. To accommodate the reviewer's question, we have included a sentence (line 178-181).

Line 263: If the sphincters are bottlenecks to capillary perfusion, how can the downstream capillaries regulate flow? Does the regulation of flow by capillaries occur only in the portion of the vascular network devoid of sphincters?

Generally, flow resistance in vascular networks is not located at specific points, but is spread across capillaries, arterioles and arteries representing a network of serial and parallel resistors (see section on the computer modeling). Sphincters represent bottlenecks that help reduce the pressure and flow into downstream capillaries to acceptable (healthy) levels. Yet, this property does not preclude the broad distribution of flow resistance and hence, the capacity of exercising flow regulation at other parts of the network, e.g. the downstream capillary or the upstream PA, remains. Note for example that both capillaries, sphincter, and the PA dilate during functional stimulation. Similarly, any constriction of downstream capillaries or upstream PA will contribute to decreased capillary blood flow irrespective of the presence of a sphincter. Along the same lines, capillary flow regulation is not only limited to capillaries devoid of sphincters.

It is probably better to think of the role of the sphincter as a facilitator. The sphincter facilitates *comparable conditions* among the capillary networks branching off the PA at different cortical depths, i.e. despite each branch having different inlet pressure. Hence, we believe that the regulatory roles of downstream capillary segments are similar in capillaries with and without a sphincter.

Figure 2: The designation of "positive" is confusing. On how many vessels and in how many mice was the assessment of number and location made? The "biophysics" was predicted according the Poiseuille's law and not assessed experimentally as the title of the figure implies.

We thank the reviewer for pointing out this source of confusion and have changed the wording on figure 2.

Also, we have put in the number of mice in the figure legend and changed the title to reflect the mostly experimental data presented. The “biophysics” wording has been removed from the figure-title.

Reviewer #2 (Remarks to the Author):

The manuscript has been improved. Authors have now included a mathematical model to support their conclusion. However, panel g (Fig. 6) must be clarified. How the model can estimate sphincter contraction? Unfortunately, there seems to be a typo here, there is no panel g in Fig. 6; therefore, we do not know where to clarify.

The model can estimate both sphincter dilation and contraction at steady state, by 1) entering the observed (or chosen) vessel segment diameters, including sphincters, into the model, 2) calculating the vessel segment resistances, and 3) solving for flow. This is basically what we do along the x-axis in Fig. 6b-c.

Figure 2, which has been barely modified, is still unclear. How many PA have been analyzed, how many sphincters have been counted. These information are not in the data source. On panel g (left), characters are too small, what does the Y axis mean? Please clarify the legend.

Thank you for the comment. We have updated figure 2 with new graph titles, new axis titles (with details in the legend) and larger font size. We have also added exactly how many PAs were analyzed and how many branchpoints in total, in addition to the total number of mice. We have also added a sentence about the average percentage of precapillary sphincters per PA and percentage of PAs with at least one sphincter (line 65-67).

Reviewer #3 (Remarks to the Author):

Grubb and colleagues have provided a thorough response letter to the reviewers' comments and have substantially modified the manuscript. I believe these revisions have significantly improved the manuscript. The authors have addressed all of my concerns. I only have a couple of minor comments to address:

1. Extended data Table 1 now has the n-number for vessels from N animals. However, the difference between n and N are not actually described in the table or its legend. Please include n = vessels and N = animals.

Thank you for pointing that out. We have added it to the table 1 legend.

2. The new schematic in Fig 1E and Extended Data Fig 8 is excellent, and that it was drawn based on an actual image is even better. I think including the confocal image presented on p16 of the rebuttal letter should be included in Extended Data Fig 8 to show the accuracy of the schematic.

Thank you for the nice comment. We have now added the maximal intensity projections to the Extended data Fig. 8.

REVIEWERS' COMMENTS:

Reviewer #1 (Remarks to the Author):

No further comments.

Reviewer #2 (Remarks to the Author):

The authors have addressed all of my concerns.